

# Boninite and boninite-series volcanics in northern Zambales ophiolite: Doubly-vergent subduction initiation along Philippine Sea Plate margins

Americus Perez[1], Susumu Umino[1], Graciano P. Yumul Jr.[2], Osamu Ishizuka[3,4]

[1]Division of Natural System, Graduate School of Natural Science and Technology, Kanazawa University, Kakuma-machi, Kanazawa, 920-1192, Japan
[2]Apex Mining Company Inc., Ortigas Center, Pasig City, 1605, Philippines
[3]Research Institute of Earthquake and Volcano Geology, Geological Survey of Japan, AIST, Tsukuba Central 7, 1-1-1 Higashi, Tsukuba, Ibaraki 305-8567, Japan
[4]Research and Development Center for Ocean Drilling Science, JAMSTEC, 2-15 Natsushima, Yokosuka, Kanagawa 237-0061, Japan

*Correspondence to*: Americus Perez (adperez@stu.kanazawa-u.ac.jp; americus.perez@gmail.com)

**Abstract.** A key component of subduction initiation rock suites is boninite, a high-magnesium andesite that is uniquely predominant in Western Pacific forearc terranes and in select Tethyan ophiolites such as Oman and Troodos. We report the discovery of low-calcium, high-silica boninite in the middle Eocene Zambales ophiolite (Luzon island, Philippines). Olivine-orthopyroxene microphyric high-silica boninite, olivine-clinopyroxene-phyric low-silica boninite and boninitic basalt occur as lapilli fall deposits and pillow lava flows in the upper volcanic unit of the juvenile arc section (Barlo locality, Acoje Block) of Zambales ophiolite. This upper volcanic unit in turn overlies a lower volcanic unit consisting of basaltic andesite, andesite to dacitic lavas and explosive eruptives (subaqueous pahoehoe and lobate sheet flows, agglutinate, and spatter deposits) forming a low-silica boninite series. The overall volcanic stratigraphy of the extrusive sequence at Barlo resembles Holes U1439 and U1442 drilled by IODP Expedition 352 in the Izu-Ogasawara (Bonin) trench slope. The presence of proto-arc basalts in Coto Block (45 Ma), boninite and boninite series volcanics in Barlo, Acoje Block (44 Ma) and simultaneous and post-boninite moderate-Fe arc tholeiites in Sual and Subic, Acoje Block (44-43 Ma) indicate that the observed subduction initiation stratigraphy in the Izu-Ogasawara-Mariana forearc is present in Zambales ophiolite as well. Paleolatitudes derived from tilt-corrected sites in the Acoje Block place the juvenile arc of northern Zambales ophiolite in the western margin of the Philippine Sea Plate. In this scenario, the origin of Philippine Sea Plate boninites (IBM and Zambales) would be in a doubly-vergent subduction initiation setting.

## 1 Introduction

As the surface manifestation of a convecting mantle, subduction zones play a fundamental role in material transfer from the surface to the deep interior. Early in Earth's history, the transition from a stagnant lid regime to the present mode dominated by plate forces necessitates subduction initiation. Numerical models show that plate tectonics or mantle overturn in terrestrial



planets can be triggered by narrow plumes impinging on weakened crust inducing gravitational instability and slab descent (Crameri and Tackley, 2016; Gerya et al., 2015). Internally driven mechanisms such as plume-induced subduction initiation, however, are fundamentally different from end-member models of Cenozoic subduction initiation that rely on pre-existing zones of weakness such as fracture zones and transform faults or require lithospheric collapse (Stern, 2004). How subduction

begins in the oceanic domain is one fundamental issue yet to be fully deciphered, thus is regarded as a high priority scientific target (Challenge No. 11) of the International Ocean Discovery Program (IODP) for 2013 to 2023. Understanding this volcanic and tectonic process is hampered by its transient nature, with few examples in the geologic record including submarine active and paleo-forearcs and on land as supra-subduction ophiolites.

In testing subduction initiation models, the Izu-Ogasawara (Bonin)-Mariana (IBM) forearc region remains as the most

appropriate locality having a temporally distinct volcanic record of oceanic accretion and juvenile arc magmatism immediately following the onset of subduction. After the subduction of Pacific Plate beneath the Philippine Sea Plate commenced at 52 Ma, magmatism progressed from 52-48 Ma mid-ocean ridge basalt (MORB)-like proto-arc basalts (or forearc basalts) to 48-44 Ma boninite and boninite-series volcanics followed by 45-35 Ma arc tholeiites and calc-alkaline lavas (Ishizuka et al., 2011; Kanayama et al., 2012; Reagan et al., 2010). The widespread occurrence of subduction initiation

rock suites along the eastern margin of the Philippine Sea Plate is summarized in Fig.1a. Basalts of similar composition and age range to proto-arc basalts dredged along the Mariana and Bonin trench slopes were found at IODP Expedition 351 Site U1438 in the Amami Sankaku Basin (Arculus et al., 2015; Ishizuka et al., 2018). This indicates that the Oligocene Kyushu-Palau Ridge (KPR) intra-oceanic arc is built upon an igneous basement that formed part of the earliest seafloor spreading event related to subduction initiation. The subsequent late Oligocene opening of the Shikoku-Parace Vela Basin resulted in

its present configuration with Site U1438 in the backarc side of KPR. Based on dredging and dive observations in the forearc trench slope, proto-arc basalts are found at deeper depths relative to boninites which occur upslope and subaerially at Chichijima (type locality) and Mukojima island groups (Ishizuka et al., 2011; Kanayama et al., 2012; Umino & Nakano, 2007). The hypothesis that proto-arc basalt lies stratigraphically below boninite has been verified by IODP Expedition 352 based on four sites drilled in the trench-side slope of the Bonin Ridge (Reagan et al., 2015). Both spontaneous and induced

initiation have been replicated in numerical models of incipient subduction at the IBM. Earlier models with imposed plate velocities generally focused on initiation across a pre-existing weak zone, and combined with melting models could reproduce the observed proto-arc basalt–boninite stratigraphy (Hall et al., 2003; Leng et al., 2012). Subsequently, spontaneous subduction initiation has been shown to be plausible; with compositional density contrast being provided by the thickened middle crust of relic Cretaceous arc terranes (Leng and Gurnis, 2015). So far, neither spontaneous nor induced

subduction initiation models can unequivocally be attributed to the IBM forearc and the mechanism remains elusive (Keenan and Encarnación, 2016).

Here an alternative approach to exploring subduction initiation processes in the Western Pacific region is presented. Following Stern et al. (2012), this study focuses on the middle Eocene Zambales ophiolite in the Philippines as it offers ~3,200 km$^2$ exposure of Eocene supra-subduction oceanic lithosphere. Most of the Jurassic to Cretaceous ophiolites that



form basement complexes in eastern Philippines are increasingly recognized as complementary features to Cretaceous terranes in the northern Philippine Sea Plate (e.g. Amami plateau, Daito Ridge) possibly sharing a common history as the overriding plate prior to initiation of subduction (Deschamps and Lallemand, 2002; Lallemand, 2016). Therefore, juxtaposed Eocene ophiolites are potential targets to investigate subduction initiation as a plate-scale process. Regional tectonic reconstructions predict the interaction of a pre-Eocene ocean basin located east of Eurasia and the western margin of the Philippine Sea Plate (Wu et al., 2016). Coincidentally, aside from the IBM forearc region, the Dasol-Barlo locality in northern Zambales ophiolite is where middle Eocene boninitic rocks are also reported (Evans et al., 1991; Florendo & Hawkins, 1992; Hawkins & Evans, 1983). The significance of Zambales ophiolite as an analogue of the IBM forearc has been recognized since the early 1990s (Pearce et al., 1992) yet this connection remains unexplored. Motivated by how the ophiolite concept progressed through studies of the IBM forearc, we present field, petrologic, and geochemical characterization of a juvenile arc section in northern Zambales ophiolite to evaluate subduction initiation and boninite petrogenesis in a regional geodynamic context.

## 2 Geologic background

Located in western Luzon, the structurally coherent Zambales ophiolite is a representative supra-subduction ophiolite of the Western Pacific and the largest in the Philippines (Fig. 1b). The ophiolite spans the entire Zambales mountain range from 16° N and 14.7° N where it partly underlies stratovolcanoes of the Luzon Arc including Mt. Pinatubo. It consists of three structurally bounded massifs, namely Masinloc, Cabangan and San Antonio, that each preserve complete ophiolite internal stratigraphy. The ophiolite is flanked by the Central Valley Basin (CVB) to the east and West Luzon Basin to the west. Late Eocene to late Miocene Aksitero and Moriones Formations of the CVB record the transition from a hemipelagic, deep marine setting to shallower environment dominated by ophiolite-derived sedimentation signaling the uplift of the ophiolite. While sedimentary sequences west of the ophiolite, Cabaluan and Sta. Cruz Formations, indicate that terrane docking and large-scale uplift have transpired by the Pliocene (Karig et al., 1986; Schweller et al., 1984). Sporadic chert slivers with late Jurassic-early Cretaceous radiolarian fauna are found in the westernmost margin of the ophiolite (Queaño et al., 2017). Similar Mesozoic cherts associated with disrupted ophiolitic fragments in northern Luzon highlights the existence of a buried shear zone (West Luzon shear zone) west of the ophiolite (Karig, 1983; Queaño et al., 2017).

Petrologic, geochemical and isotopic studies reveal that Zambales ophiolite represents two distinct mantle-crust sequences – the 45.1 Ma Coto Block with transitional MORB affinity and 44.2 - 43.7 Ma Acoje Block with island arc characteristics (Encarnación et al., 1999; Encarnación et al., 1993; Geary et al., 1989; Hawkins & Evans, 1983; Yumul, 1990). The existence of a NNE-SSW trending structural boundary previously characterized by Hawkins and Evans (1983) as a left-lateral fault between Acoje and Coto Blocks within the Masinloc massif is supported by gravity and magnetic data with anomaly contrasts which extend at depth (Salapare et al., 2015). Although the juvenile arc character of Acoje Block and the supra-subduction origin of Zambales ophiolite are clear, the crustal nature and tectonic environment of Acoje and Coto



blocks remain unresolved. Acoje Block as defined by Yumul et al. (1990) consists of the northern Masinloc massif and the San Antonio massif while the Coto Block consists of the southern Masinloc massif and the Cabangan massif (Fig. 1b). The present disposition of Acoje Block is postulated to be the result of southward translation of San Antonio massif with respect to northern Masinloc massif (Yumul et al., 1998). Crustal thickness estimates, based on SW-NE transects of the northern

Masinloc massif, are up to 9.5 km for the mantle section and 7 km for the lower crustal section. The fertile to moderately depleted Acoje mantle section is comprised of harzburgite and lherzolite with spinel Cr# [Cr/(Cr+Al)] ranging from 0.18 to 0.56, generally increasing towards the mantle-crust transition zone. Harzburgites from the uppermost mantle section are strongly depleted in light and middle rare earth elements (LREEs and MREEs) with equilibration temperature and oxygen fugacity estimates of 730° C and 1.9 $\Delta$log(fO$_2$) units above the FMQ buffer, respectively (Evans & Hawkins, 1989; Tamayo,

2001; Yumul, 1989). From the residual mantle section, the mantle-crust transition zone passes through interlayered ultramafic cumulates and dunite to layered cumulate gabbronorite. The Acoje transition zone hosts podiform chromite deposits (Cr# = 0.71-0.77) and Ni-Cu sulfides both of which are PGE-bearing (Bacuta et al., 1990). Refractory transition zone dunites, characterized by spinel Cr# greater than 0.6, are interpreted to be of cumulate origin (Abrajano et al., 1989). The Acoje Block lower crustal section is dominated by cumulate gabbronorites with olivine-clinopyroxene-orthopyroxene-

plagioclase crystallization sequence. Coexisting plagioclase and clinopyroxene in gabbronorite are calcic (anorthite content [Ca/(Ca+Na+K)] = 89-94) and magnesian (Mg# [Mg/(Mg+Fe)] = 0.80-0.87). In addition to petrological criteria discussed above, the arc affinity of Acoje Block has been demonstrated using immobile element-based geochemical fingerprinting of volcanic and hypabyssal rocks, as well as constituent clinopyroxene (Yumul, 1996). On a regional scale, the ophiolite appears to have a domal structure with a north-south trending axis. The volcanic section of Acoje Block in northern

Masinloc massif dips northwest and is unconformably overlain by middle to late Miocene sedimentary sequences.

## 3. Volcanic geology

Based on observations around the Barlo massive sulfide mine, two volcanic units with distinct lithofacies are distinguished in the Acoje Block extrusive sequence—a dominantly pyroclastic lower basaltic andesite-andesite unit overlain by an upper boninite unit with minor boninitic basalts. Structural measurements of bedding planes in 136 locations constrain the gross

structure of the crustal sequence, consisting of a NE-SW trending doubly-plunging anticline forming a dome structure and NW-plunging anticline (Fig. 2). Estimated thickness of the upper unit and lower volcanic units are 680 and 520 meters, respectively.

South of Barlo mine, boninite dike swarms with conjugate intrusive directions change into a succession of submarine explosive volcanic deposits. In order of ascending stratigraphic level, the lower basaltic andesite-andesite unit consists of tuff

breccia with block-sized pyroclasts, subaqueous pahoehoe lava flows with ropy wrinkles and stretched vesicles on lobe crust (Umino et al., 2002; Umino et al., 2000) (Fig. 3f), moderately welded agglutinate, welded scoria and spatter deposits marginal to a NW-SE trending fissure vent, and an uppermost fall-out deposit of glassy lapilli tuff (Fig. 3e). In places, tuff



breccia is cut by NE-dipping boninite dikes. The dominant occurrence of flat pahoehoe lobes suggest the pre-existence of subhorizontal topography (Umino et al., 2002). The volcanic facies recognized in the upper section of the lower volcanic unit (agglutinate, scoria and spatter deposits surrounding fissure vent) are characteristic of intermittent Strombolian to Hawaiian fire fountaining.

5 NW-dipping boninite tuff breccia and pillow lavas of the upper unit (Fig. 3b) overlie hyaloclastite, spatter deposits and pillow lavas (Fig. 3c) of the lower basaltic andesite-andesite unit at an exposure north of Mt. Sol. A sill directly above lower unit pillow lavas has a whole-rock K-Ar age of 44.1 ± 3.0 Ma (Fuller et al., 1989). Pillow lavas of the upper and lower unit show distinct morphological features based on vertical (V) and horizontal (H) axes measurements after the method of Walker (1992). Pillow lavas of the lower unit are marked by lower aspect ratios (n=92, median H=0.53 m, median V=0.34 10 m) compared to boninitic basalts of the upper unit which are elongate (n=15, median H=0.74 m, median V=0.32 m). SW of Barlo mine, NE-dipping pillow lavas of the upper unit are overlain by middle-late Eocene radiolarian-bearing claystone (Schweller et al., 1984). Further southwest in a synclinal area with upright structures preserved, a ~30 meter-high boninite pillow volcano is recognized (Fig. 3d). It consists of flattened flow lobes in the summit with pillow lavas dipping downslope along its flanks. This volcanic construct is one of several discrete boninitic volcanic edifices rising above the lower unit. 15 Elsewhere, these pillow lavas are cut by SW-dipping dikes with glassy margins and overlain by glassy boninite lapilli tuff fall deposit (Fig. 3a).

## 4 Sampling and analytical methods

A total of one hundred fifty-two (152) samples were collected during the 2016 mapping campaign. For this study, a subset of forty-four (44) samples located along NW-SE transects were selected for whole rock geochemical analyses and screened 20 through visual and microscopic assessment of secondary alteration. Location of samples is shown in the geologic map in Fig. 2. Rock slabs were cut to remove altered surfaces. Initially, slabs were crushed coarsely to manually separate low temperature secondary minerals such as quartz and calcite. Coarse crushing step was followed by rinsing with deionized water and drying in an oven at 110 °C for at least 12 hours. Iron mortar and agate mill were used for fine crushing and grinding, respectively. Ignition loss is taken as the normalized lost weight after ignition of ground rock powder at 900 °C for 25 four hours. Major element compositions were determined by x-ray fluorescence (XRF) spectrometry using a PANanalytical Axios spectrometer at the Geological Survey of Japan. Glass beads were fused using a Tokyo Kagaku bead sampler (TK-4500) at 1:10 dilution ratio as mixtures of 0.5 g (±0.0002) rock powder and 5 g (±0.001) lithium tetraborate alkali flux ($Li_2B_4O_7$, Merck Spectromelt A10). Precision, measured as percent relative standard deviation (%RSD), is better than 2% based on repeated measurements of JB-2, BIR-1 and BHVO-2; accuracy is better than 2% as well. Trace element (REE, V, 30 Cr, Ni, Li, Be, Rb, Sr, Y, Zr, Nb, Cs, Ba, Hf, Ta, Pb, Th and U) concentrations were determined by inductively coupled plasma mass spectrometry (ICP-MS) using an Agilent 7900 instrument also at GSJ/AIST. Samples were digested for at least 48 hours on a hotplate using screw top Teflon beakers with a mixture of HF and HNO3 at 5:1 ratio. Dissolution procedures





were performed in a Class 1000 clean laboratory. 12 international standards (BIR-1, BRR-1, JB-2, JB-3, JA-1, JGb-1, JA-2, BCR-1, AGV-1, JB-1a, BHVO-2 and BE-N-1) were used to construct calibration lines. JB-2 and JB-3 solutions at similar dilution levels were used as external standards. Reproducibility is better than 2 % for REEs and better than 3 % for the rest of the trace elements except for Ta (6.5 %) and Be (3.6 %). Mean percent error is generally within 5–10 % relative to the

preferred values of Jochum et al. (2016) and Dulski (2001).

Major element compositions of constituent mineral phases were determined with a JEOL-8800R electron microprobe at Kanazawa University using 3 μm probe diameter, 20 nA probe current and 20 kV accelerating voltage. Natural (Kurose olivine and clinopyroxene) and synthetic mineral standards were used for calibration and data were corrected using the ZAF method.

## 5 Results

Representative mineral chemistry data are listed in Supplementary Table 1. Results of whole-rock major and trace element analyses are given in Supplementary Table 2 with total Fe represented as $Fe_2O_3$. In succeeding discussions, all samples are normalized to 100 % volatile free with total Fe recalculated as FeO* to enable comparison with compiled datasets.

### 5.1 Textures and phase chemistry

Zambales high-silica boninite (as defined in 5.2) is mostly aphyric and consists of subhedral olivine microphenocrysts (Mg# [Mg/ (Mg + $Fe^{2+}$)] = 0.88–0.91), abundant elongate enstatite microphenocrysts (Mg# = 0.86–0.89, $Wo_{2.6–4.9}En_{82–86}$) with augite ± pigeonite overgrowth (Mg# = 0.67–0.84, $Wo_{31-44}En_{34-57}$), and chromian spinel (Cr# [Cr/(Cr+Al)] = 0.69–0.84, Mg# =0.53–0.69) set in a glassy groundmass with feathery quench clinopyroxene (Figs. 4a-b). This assemblage corresponds to Type II boninite of Umino (1986) in samples described from the type locality. Unlike low-Ca, high-silica boninites from

Ogasawara, clinoenstatite has not been identified so far. Comparative mineral chemistry of Zambales and Ogasawara boninite is shown in Fig. 5. Olivine, orthopyroxene and spinel compositions of Zambales and Ogasawara boninite are almost identical except for a limited range of olivine Mg#, which is most likely due to a small number of samples analyzed (two for olivine and four for spinel), and lower spinel Cr# in Zambales boninite (Taylor et al., 1994; Umino, 1986; Yajima & Fujimaki, 2001). NiO contents of olivine range from 0.18–0.41 wt% and plot on fractionation trends from the mantle olivine

array. Subhedral olivine microphenocrysts in high-silica boninite (ZM2-100A) is zoned with magnesian cores. The range of spinel Cr# of Zambales boninite overlap with Acoje podiform chromitites and transition zone dunites. Enstatite microphenocrysts with sieve texture and reverse zoning are recognized together with embayed quartz xenocrysts (Figs. 4c-e). Reversely zoned enstatite has oscillatory-zoned magnesian rims (Mg# = 0.86–0.87) with resorbed and dissolved cores (Mg# = 0.68–0.77). Similar xenocrystic quartz and sieve-textured enstatite have also been noted in Ogasawara boninite

(Watanabe and Kuroda, 2000). Glassy olivine-phyric and olivine-clinopyroxene-phyric low-silica boninite varieties are also



present (Fig. 4f). Hypocrystalline boninite pillow lavas consist of olivine, usually altered and replaced by chlorite and carbonate minerals, subhedral clinopyroxene phenocrysts, elongate orthopyroxene and plagioclase microphenocrysts. Highly-phyric boninitic basalt has a phenocryst assemblage of olivine replaced by calcite, zeolite and clay minerals, euhedral to subhedral augite (Mg# = 0.82–0.90, $Wo_{41-43}En_{48-53}$) with oscillatory and sector zoning, and rare corroded enstatite (Mg# = 0.85–0.88, $Wo_3En_{82-84}$) with reverse zoning. Chromian spinel (Cr# = 0.62–0.71) occurs near phenocryst margins and as inclusions in olivine. Peculiar zoning relationship of augite and enstatite demonstrates that some pyroxenes are xenocrysts in nature. Groundmass consists of < 0.3 mm long plagioclase microlites with swallow-tail terminations, spherulitic clinopyroxene and altered interstitial glass totally replaced by clay minerals. Acicular zeolite occurs as vesicle fillings. We note that such textures have been described in samples reported as boninite by Evans et al. (1991).

Lower-unit basaltic andesite and andesite pillows and lapili tuff are marked by the presence of abundant plagioclase laths as glomerocrysts and sparse cpx ± opx microphenocrysts with two-pyroxene intergrowths in a glassy groundmass with plagioclase microlites and magnetite (Fig. 4g).

## 5.2 Whole-rock geochemistry

The Acoje Block extrusive section at Barlo is composed predominantly of basaltic andesite and andesite. Loss on ignition (LOI) values are generally between 4–7 wt% and total alkalis are < 3 wt%. Cs, Rb, K, Li, Ba and Mn in some dikes and pillow lavas, especially those collected nearest to the massive sulfide mine, are highly scattered and plot away from general fractionation trends shown by the majority of samples. The rest of the samples show variations in the major element and trace element compositions consistent with fractional crystallization and are deemed as primary. Pillow lava, tuff breccia and lapilli tuff of the upper volcanic unit and dikes from the uppermost section of the dike-sill complex are primitive with > 8 wt% MgO and < 0.5 wt% $TiO_2$, satisfying the IUGS criteria for boninite (Le Maitre, 2002). Pillow lavas that qualify as boninite based on MgO and $TiO_2$ contents but with < 52 wt% $SiO_2$ are classified as boninitic basalts. Following the classification of Kanayama et al. (2013) and Reagan et al. (2015) both high-silica (HSB) and low-silica (LSB) boninite subtypes are recognized (Fig. 6). This compositional division is consistent with petrographic observations, with HSB having olivine + orthopyroxene microphenocrysts and LSB with olivine and olivine + clinopyroxene phenocrysts. Boninitic samples previously described at Barlo are actually from the dike section and lower section of the upper unit and with sampling limited to Balincaguin River and the periphery of Barlo VMS mine (Evans et al., 1991; Hawkins & Evans, 1983; Tamayo, 2001; Yumul, 1990). Besides the low-silica boninite and boninitic basalt samples from previous studies with altered equivalents having >3.5 wt % $Na_2O+K_2O$, 3–5 wt % LOI and possibly up to 4 wt% $SiO_2$ enrichment, this is the first report of pristine high-silica boninite in Zambales ophiolite. Boninite from Zambales (n=14) have $CaO/Al_2O_3$ ratios less than 0.76, mostly within 0.63–0.71 with FeO*/MgO ratios <1 (Mg# values from 63 to 73). Ni and Cr contents are high, ranging from 72 μg/g to 292 μg/g and 145 μg/g to 727 μg/g, respectively. Boninitic basalts (n=3) are primitive as well with 12.4–13.6 wt% MgO and 712–1113 μg/g Cr.



Pillow lava and lapilli tuff of the lower volcanic unit and dikes from the deeper section of the dike-sill complex are andesitic in composition with 55.5–63.5 wt% $SiO_2$, 2.4–7.1 wt% MgO and 0.31–0.47 wt% $TiO_2$ (Fig. 6). These andesites (n=26) are magnesian with Mg# mostly within 0.47 to 0.54, $CaO/Al_2O_3$ ratios < 0.63 and FeO*/MgO ratios > 1.16. Ni and Cr contents are low at 8–58 μg/g and 3–47 μg/g. MgO and $SiO_2$ compositions of lower unit volcanics and corresponding dikes lie above

the island arc basalt (B)-basaltic andesite (BA)-andesite (A)-dacite (D) discriminant line of Pearce and Robinson (2010) and follow a curvilinear fractionation trend from low-silica boninite; thus, is recognized here as forming a low-silica boninite differentiation series (Fig. 6). Dacitic pillow lavas represent the differentiated end of the LSB series. A gap between andesitic boninite-series volcanics with < 60 wt% $SiO_2$ and dacite (62–63 wt%) is present. A single dike (B-47 d4) that cross cuts the LSB-series dikes is the only sample with > 0.5 wt % $TiO_2$. Although the pervasive clay alteration of samples from

the Barlo massive sulfide mine hinders the direct characterization of the host rocks, the massive sulfide deposit is most likely hosted by the lower volcanic unit and possibly up to the lowermost section of the upper unit based on stratigraphic relationships and comparison of field and core descriptions (Paringit, 1977). The occurrence of massive sulfide-bearing horizons at Barlo is also consistent with massive sulfide mineralization in Oman, Troodos and Bonin Ridge which occur either below or within the boninitic volcanic section (Ishizuka et al., 2014; Umino et al., 2009).

Trace element compositions of Zambales boninite and boninite series volcanics are marked by low abundances relative to MORB (Fig. 7). Extended trace element patterns arranged after Pearce and Parkinson (1993) show the dominance of slab-derived components with notable enrichment in fluid-mobile elements (Cs, Rb, Ba, U, K, Pb, Sr, Li, Na) and depletion of high field strength elements (Nb, Ta). Hf shows a positive anomaly relative to Sm while Ti shows a negative anomaly relative to Y. In contrast with samples from Ogasawara that exhibit characteristic U-shaped chondrite-normalized rare earth

element (REE) patterns, Zambales boninite and boninite-series volcanics exhibit spoon-shaped REE patterns with heavy REEs (HREE) at 5–15 times chondrite values and weak negative Eu anomaly (Eu/Eu* = 0.9). Boninitic basalts are more depleted in middle and heavy REEs with just 3–5 times the chondrite values. Excluding moderately altered samples, chondrite normalized La/Ce ($[La/Ce]_N$) ratios of Zambales and Ogasawara boninite are both <1. The range of $[Sm/Yb]_N$ ratios, as measure of middle REE depletion relative to heavy REE, however, are distinct at 0.06–0.09 for Zambales and

0.02–0.05 for Ogasawara. The effect of fractionation in Zambales samples is shown as increasing trace element abundances relative to a glassy olivine-phyric LSB (ZM2-107) with 12.4 wt% MgO. The evolved dacitic samples does not differ significantly with just twice the normalized REE abundances of the most primitive sample. Trace element pattern of a single dike (B-47 d4) with > 0.5 wt% $TiO_2$ is visibly different with muted fluid-mobile element enrichment without a negative Ti anomaly, and smooth, concave downward LREE-depletion (Fig. 7).

The volcanic sequence at Barlo, based on observed local stratigraphic relationships, is HSB-LSB-boninitic basalt-LSB series volcanics in descending order (Fig. 3). This volcanic stratigraphy is remarkably similar to Holes U1439 and U1442 of IODP Exp. 352. Both the Barlo extrusive section and Holes U1439 and U1442 are marked by abundance of LSB relative to HSB (Reagan et al., 2015). In contrast, the onshore boninite sequence at Ogasawara is dominated by 48–46 Ma HSB and HSB series volcanics interbedded with minor LSB (Maruberiwan Formation), which are overlain by 45 Ma LSB and calc-alkaline





andesite-dacite (Mikazukiyama Formation) (Kanayama et al., 2012; Taylor et al., 1994; Umino, 1986). Several dikes in the Subic crustal section in the southern San Antonio massif have compositions like Barlo boninitic basalts (Yumul et al., 2000), albeit their petrographic characteristics have not been described. This raises the possibility that boninite is present in other volcanic sections of the Acoje Block as well. The rest of the volcanic sections of the Zambales ophiolite (Coto volcanic

section in Tarlac, Sual dikes north of Barlo, and Subic dikes and pillow lavas in the south) are dominated by basalts and basaltic andesites with > 0.5 wt% $TiO_2$ that can be classified as moderate-Fe tholeiites. The tholeiite dike (B-47 d4) that cross cuts low-silica boninite dikes in the deeper dike swarm section at Barlo is similar in composition and correlatable with 44.0±3.0 Ma dikes in the Sual crustal section located NE of Barlo, suggesting that tholeiites are coeval or slightly younger than the Barlo extrusive section.

**6 Discussion**

**6.1 Low-pressure fractionation and magma mixing**

Early experimental studies of primitive boninite produced andesitic primary liquids and led to the initial characterization of boninite as a primary magma (Kuroda et al., 1978). Subsequent detailed petrologic analyses resulted in the recognition of the importance of low-pressure fractionation and magma mixing processes to account for compositional variation of boninite

and boninite series volcanics (Dobson et al., 2006; Taylor et al., 1994; Umino, 1986). Meijer (1980) introduced the concept of the boninite series to describe cogenetic bronzite andesite, dacite and rhyolite formed through differentiation of boninite, analogous to tholeiitic and calc-alkaline magma series. Figure 6 shows oxide vs. $SiO_2$ variation diagrams of Zambales boninite and boninite series volcanics. Also shown are boninite and boninite series volcanics from Ogasawara (Kanayama et al., 2012; Nagaishi, 2008; Taylor et al., 1994; Umino & Nakano, 2007; Yajima & Fujimaki, 2001) and Holes U1439 and

U1442 of IODP Expedition 352 (Haugen, 2017; Reagan et al., 2015). MgO, $TiO_2$, CaO and $Al_2O_3$ contents of HSB and LSB form subparallel trends suggesting distinct parental magma compositions and fractionation paths. Parental HSB is characterized by higher MgO, lower $TiO_2$, $Al_2O_3$ and CaO at a given $SiO_2$ than LSB. HSB is marked by the abundance of orthopyroxene and, in some localities, the presence of clinoenstatite. Umino (1986) describes the crystallization sequence at Chichijima as chromian spinel, olivine, clinoenstatite, enstatite and augite. HSB differentiation series is best exemplified by

samples from Ogasawara and Site 786, ODP Leg 125 (Arculus et al., 1992). Boninite from other notable Western Pacific localities such as the Mariana Trench (Bloomer & Hawkins, 1987; Dietrich et al., 1978), Cape Vogel, Papua New Guinea (Jenner, 1981; König et al., 2010), Nepoui, New Caledonia (Cameron, 1989; Cluzel et al., 2016) and Hahajima seamount (Li et al., 2013) are all high-silica sub type. Zambales HSB show limited compositional variation with most of the samples collected in an area north of Mt. Sol, possibly from a single volcanic edifice. On the other hand, LSB crystallization

sequence is controlled by olivine and clinopyroxene fractionation (Kostopoulos and Murton, 1992; Natland, 1982). LSB differentiation series is typified by suites from northern Zambales, Holes U1439 and U1442 of Expedition 352 and Hole 458, ODP Leg 60 (Meijer, 1982; Sharaskin, 1982). Other LSB and LSB series localities include Facpi Formation in Guam





(Reagan & Meijer, 1984), Tonga Trench (Falloon et al., 2007; Falloon and Crawford, 1991), Troodos ophiolite (Cameron, 1985; König et al., 2008; Osozawa et al., 2012; Pearce & Robinson, 2010) and Oman ophiolite (Ishikawa et al., 2002; Kusano et al., 2017; Kusano et al., 2014; Nagaishi, 2008).

Using spinel-hosted melt inclusions reported by Umino et al., (2017) as parental magma compositions, isobaric fractional crystallization paths of residual liquids at +1 Δlog(FMQ) are modelled using rhyoliteMELTS v.1.2.0 (Ghiorso and Gualda, 2015). The compositions of primary spinel-hosted boninitic melt inclusions from Ogasawara (2.1–3.4 wt% $H_2O$) and Guam (3.8 wt% $H_2O$) are in accord with experimental glasses formed by melting synthetic Troodos harzburgite and depleted Tinaquillo lherzolite at 1.5 GPa (Falloon and Danyushevsky, 2000) (Fig. 6). The range of whole-rock HSB compositions can be reproduced by low pressure (1–2 kb) fractional crystallization of parental Ogasawara HSB magmas at lower water content (1 wt%). $TiO_2$ and CaO contents of Zambales HSB are in the upper end of Ogasawara HSB compositions and alternatively can be produced by fractionation from a parental composition straddling between the HSB and LSB boundary of Reagan et al. (2015). Given a primary LSB magma composition identical to a spinel-hosted melt inclusion from Guam, low pressure (1–2 kb) fractional crystallization at lower water content (1 wt%) can likewise account for the major element variation of Zambales LSB and LSB series volcanics. The modelled early crystallization of cpx relative to opx and the onset of plagioclase crystallization following cpx at ~57 wt% $SiO_2$ is reflected in the modal compositions of Zambales LSB and LSB-series volcanics (Fig. 4). The occurrence of differentiated LSB series volcanics in the base of the volcanic pile suggests that initial LSB magmas underwent protracted crystal fractionation in magma chambers at depth. Influx of primitive HSB perturbed sub-volcanic magma chambers and likely induced magma mixing. The presence of reversely zoned enstatite microphenocrysts in Zambales HSB, attest to high-level magma mixing between an olivine+orthopyroxene saturated HSB magma and a pre-existing LSB series andesitic magma. Zambales HSB slightly deviate from the modelled Ogasawara HSB fractionation paths and lie on a mixing line with potential HSB and LSB-series end member components having $SiO_2$ contents that does not vary significantly at 58 and 57 wt % respectively.

The markedly divergent fractionation paths followed by boninite and boninite series volcanics from Barlo and basalts from various volcanic sections of the Zambales ophiolite are shown in the MgO, $TiO_2$ and FeO*/MgO vs $SiO_2$ variation diagrams (Fig. 6). Zambales LSB and LSB series volcanics follow fractionation paths of hydrous primitive arc magmas (Grove et al., 2012). Moderate-Fe tholeiitic basalts and basaltic andesites from the rest of the volcanic sections of Zambales ophiolite (Evans et al., 1991; Geary et al., 1989; Hawkins & Evans, 1983; Tamayo, 2001; Yumul, 1990) and proto-arc basalts from the Izu-Ogasawara-Mariana forearc (Arculus et al., 2015; Ishizuka et al., 2011; Reagan et al., 2010, 2015) exhibit large FeO*/MgO ratio variation at a restricted $SiO_2$ interval and define a slope steeper than the discriminant line of Miyashiro, (1974).

## 6.2 Shallow slab contributions to a less depleted mantle source

Although there is a general consensus on boninite petrogenesis (Crawford et al., 1989; Pearce et al., 1992), the requisite sub-arc conditions ideal for boninite generation span a range that may involve heterogeneous mantle wedge sources (depleted



lherzolite to harzburgite) and addition of diverse slab-derived components. We highlight the peculiar trace element characteristics of Zambales boninite; geochemical modelling will be presented elsewhere. Fig. 7 shows MORB and chondrite-normalized trace element patterns of Zambales boninite and boninite-series volcanics together with boninite suites from Ogasawara, Holes U1439 and U1442 of IODP Expedition 352, Troodos and Oman ophiolites, Nepoui (New Caledonia) and Cape Vogel (Papua New Guinea). Boninite from Zambales, Oman and Troodos ophiolites all show spoon-shaped chondrite normalized REE patterns with heavy REEs that are 5–9 times chondrite values. Based on HREE abundances and spinel Cr#, Zambales boninite is derived from a mantle source that is less depleted than Ogasawara boninite (Fig. 7).

Superimposed on the depleted character of boninite are slab-derived components whose nature can be explored using normalized trace element diagrams and trace element ratios with varying compatibilities. The magnitude of hydrous fluid mobile element enrichment in boninite from Ogasawara, Zambales, Oman and Troodos ophiolites are comparable. LREE–enrichment is most prominent in Nepoui and Cape Vogel boninites. All boninite suites except for Nepoui and Cape Vogel have elevated Ba/Th ratios (Fig. 8a). Ba is strongly partitioned into aqueous fluids relative to Th, thus is a good proxy for the addition of subduction components released at shallow depths – sz1, Ba-only component of Pearce et al. (2005) (Keppler, 2017; Ribeiro et al., 2015). Increasing Ba/Th ratio is mirrored by Sr/Nd ratio for Zambales, Oman and Troodos boninite (Fig. 8b). Ogasawara, Cape Vogel and Nepoui follow a vertical Th/Yb vector towards sediments – the component 1 of Elliott (2003). We interpret this decoupling as an indicator of increasing contributions from a deep subduction component that is usually characterized either as sediment melt and/or siliceous fluid (Pearce et al., 2005). Zambales boninite and boninite-series volcanics have U/Th ratio (0.62–0.77) much higher than Cape Vogel boninites (0.20–0.29). High U/Th ratio is associated with saline and oxidizing fluids, while low U/Th ratio point to transport by silica-rich fluids (Keppler, 2017). Zambales and Cape Vogel boninites both have a uniform La/Th ratio of 0.2. In the La/Th vs. Sm/La diagram of Plank (2005), Zambales, Oman, and Troodos boninites lie on a mixing line between sediments with Th/La ratio of 0.30–0.35 and a mantle source more refractory than MORB with a Sm/La ratio of 1.75. Ogasawara, Cape Vogel and Nepoui boninites may lie on the mixing line between sediments with Th/La ratio of 0.1–0.2 and a highly-depleted mantle source with Sm/La greater than 2. An alternative explanation is that low Sm/La ratios of Ogasawara, Cape Vogel and Nepoui boninites dominantly reflect LREE-enrichment that masks mixing relationship. The shared characteristics of Zambales, Oman, and Troodos boninites are further exemplified by subchondritic bulk Earth Nb/Ta ratios with low Zr/Hf and Zr/Sm ratios (< 30) characteristic of depleted mantle (Münker et al., 2003; Weyer et al., 2003; Zanetti et al., 2006) (Fig. 8c). High Zr/Sm ratios (> 40) of Ogasawara, Cape Vogel and Nepoui boninites have been ascribed to the presence of residual amphibole in the slab (Cluzel et al., 2016; König et al., 2010; Pearce et al., 1992). In summary, trace element ratios show that aqueous fluids are the dominant slab components in Zambales boninite and boninite series volcanics with very minor contribution from slab melts (Umino et al., 2015). This is corroborated by enriched Pb and Sr isotopic compositions of Acoje Block uppermost crustal differentiates and cumulate ultramafic samples that do not show an inverse relationship with Nd isotopic compositions (Encarnación et al., 1999).



### 6.3 Subduction initiation origin of Zambales ophiolite

The 45 Ma crustal section of the Coto Block has been shown to be of island arc tholeiite (IAT)-transitional mid-ocean ridge basalt (MORB) composition; in immobile element-based trace element discrimination diagrams, Coto Block dikes and lavas from Tarlac are robustly characterized as distinct from Mariana Trough back-arc basin basalts (Geary et al., 1989).

Decreasing REEs, $TiO_2$, Zr and Y in Coto volcanics, dikes in the Coto mantle section to Acoje volcanics document the progressive depletion of the mantle source in Zambales mantle wedge (Yumul, 1990). Succeeding zircon U-Pb geochronology established the 1–2 Ma age difference between Coto and Acoje blocks (Encarnación et al., 1993). Thus, combined geochemical and age constraints preclude a back-arc basin origin for Coto Block as forwarded by Hawkins and Evans, (1983), and are more compatible with a proto-forearc setting as originally suggested by Geary et al., (1989).

Although Coto samples have variable Ti/V ratios (Shervais, 1982) (Fig. 6), $Ti_8$ and $Si_8$ values are comparable to proto-arc basalts from the Bonin Ridge, Mariana Trench slope, Hole U1430 of IODP Exp. 351 and Hole 1440 of IODP Exp. 352 (Arculus et al., 2015; Ishizuka et al., 2011; Reagan et al., 2010, 2015) (Fig. 9). $Si_8$ and $Ti_8$ values of Coto Block basalts are also distinct from Celebes Sea Basin and West Philippine Basin MORB (Pearce et al., 2005; Savov et al., 2006; Serri et al., 1991; Zakariadze, 1981). From observations in the field, oceanic accretion can be inferred from the Coto crustal section

based on a well-developed sheeted dike complex, in contrast to the mutually intrusive dikes of the Acoje crustal section. With the recognition of boninite and boninite-series volcanics (Barlo), simultaneous and post-boninite moderate-Fe arc tholeiites (Barlo-Sual, Subic) in the crustal section of Acoje Block, it is apparent that the subduction initiation chemostratigraphy (Ishizuka et al., 2011; Whattam & Stern, 2011) is present in Zambales ophiolite as well. Although stratigraphic relationships of individual volcanic sections may be obfuscated by emplacement-related tectonic deformation,

by considering available age constraints, the crustal sections of the Zambales ophiolite represent proto-arc accretion and juvenile arc sections supplied by separate magma plumbing systems as observed in IODP Exp. 352 sites (Reagan et al., 2015).

### 6.4 Doubly vergent subduction initiation

Previous attempts to ascertain the provenance of Zambales ophiolite focused on its characterization as a marginal basin and

thus linked it to the surrounding Eocene marginal basins such as the Celebes Sea Basin and Molucca Sea Basin. In the most comprehensive petrological and geochemical study of Philippine ophiolites, Tamayo et al. (2004) notes the strong subduction geochemical imprint of Zambales ophiolite and placed it as a marginal basin between Celebes Sea Basin and the western margin of the Philippine Sea Plate. The difficulty in reconstructing the Paleogene tectonic configuration of the southeast Asian region, specifically the spreading history of the oceanic domain east of Sundaland and west of the Philippine

Sea Plate, stems from the fact that this region has been fully consumed along convergent margins east of southeastern Eurasia (Zahirovic et al., 2014). This gap was addressed by Wu et al. (2016) through the novel use of structural unfolding of subducted slabs imaged by seismic tomography. In their preferred model (model 1b), unfolding of subducted slabs beneath





southern Eurasia reveals that the Philippine Sea Plate was bordered by Cretaceous oceanic crust (the East Asian Sea) in its western margin. Paleomagnetic data derived from sites in the Philippine Sea Plate shows northward translation concomitant with clockwise rotation (Hall et al., 1995; Richter & Ali, 2015; Yamazaki et al., 2010) (Fig. 10a). Eocene to Miocene sites from Luzon indicate similar northward migration; thus, the island can be regarded as the western promontory of the

Philippine Sea Plate (Queano et al., 2007). The subequatorial position of Luzon during the middle Eocene is further supported by paleomagnetic data from pillow lavas forming the basement of Luzon Central Cordillera (K-Ar age: 45.49±2.27 Ma) that yield a paleolatitude of 6.3±3 °S (Queaño et al., 2009; Sajona, 1995). East of Luzon and forming the greater part of Philippine Sea Plate, the West Philippine Basin (WPB) is characterized as a back-arc basin formed between opposing subduction zones (Deschamps and Lallemand, 2002). Pre-Eocene subduction south of WPB possibly generated the

Mesozoic arc terranes in the Philippine Sea Plate including Amami Plateau and Daito ridge as well as the Cretaceous ophiolites in eastern Philippines (Lallemand, 2016). Initiation of a long-lived subduction east of WPB produced the Izu-Ogasawara (Bonin)-Mariana (IBM) forearc (Ishizuka et al., 2011). It is in this context that we re-evaluate the origin of Zambales ophiolite.

Tilt-corrected inclination data from sites in the 44 Ma Acoje Block equates to equatorial paleolatitudes (Fuller et al., 1991).

Adopting southerly paleolatitudes in consonance with the modelled Luzon translation history of Wu et al., (2016), we make the case for the subduction initiation origin of Zambales ophiolite in the leading edge of the northwestward-moving, clockwise-rotating Philippine Sea Plate. In this scenario, the origin of Philippine Sea Plate boninites (IBM and Zambales) would be in a doubly-vergent subduction initiation setting (Fig. 10b). In contrast with IBM, incipient subduction that produced Zambales ophiolite proto-arc basalts, boninites and arc tholeiites within 3 Ma is rather short-lived and alludes to

rapid emplacement. The timing of subduction initiation in the western margin of Philippine Sea Plate is yet to be constrained. The late Oligocene to early Miocene onset of true basinal sedimentation in the Central Valley Basin provides a minimum docking age between Zambales ophiolite and eastern Luzon (Bachman et al., 1983). The GPlates-based reconstruction presented here, using unfolded slab data (Wu et al., 2016), is essentially similar to the Philippine Sea Plate evolutionary models of Deschamps and Lallemand (2003). Although it is beyond the scope of this study to identify the

nature of a pre-existing structural boundary as a transform fault or strike-slip extension of a subduction zone, we note that both Zambales and IBM are orthogonal to the West Philippine Basin spreading center and the Cretaceous arc terranes in the overriding plate. Contrary to a subduction initiation origin, Lallemand (2016) posits that proto-arc basalts (or forearc basalts) formed in short lived Eocene oceanic basins (e.g. Kita-Daito and Minami-Daito basins) bounded by the Cretaceous arc terranes. We believe that this might be the case for Eocene ophiolites in eastern Philippines (e.g. Angat ophiolite) but not for

Zambales ophiolite. The early Eocene age (48.1±0.5 Ma) of Angat ophiolite does not necessarily prove an affinity with Zambales as suggested by Encarnación et al. (1993), rather it simply demonstrates that basement complexes in the Philippine Mobile Belt consists of juxtaposed geochemically distinct Eocene and Cretaceous ophiolites (ophiolite belts 1B and 1C of Tamayo et al., 2004) (Fig. 1a). Slightly older Eocene ages (e.g. 50.83±3.35 Ma) were obtained by subsequent whole-rock K-Ar and SHRIMP zircon U-Pb dating of ophiolitic samples from eastern Luzon (Sajona, 1995; Tani et al.,



2015). In addition, these Cretaceous and Eocene ophiolites are associated with Eocene volcano-sedimentary arc sequences (Caraballo, Maybangain, Anawan and Payo Formations). This differs sharply with the deep-marine, isolated environment deduced from the Eocene sedimentary carapace of Zambales ophiolite.

Although current terrestrial subduction is dominantly asymmetric, it is interesting to note that two-sided subduction is what is essentially produced in 2-D and 3-D models of mantle convection (Gerya et al., 2008; Wada & King, 2015). We speculate that the location of Philippine Sea Plate (PSP) in the nexus of Pacific, Indo-Australian and Eurasian plates and their long-term Cenozoic plate motion makes doubly-vergent subduction initiation along its margins feasible. The dynamics of sustained double-vergent subduction is examined by Holt et al. (2017) but doubly-vergent subduction initiation is yet to be explored by numerical modelling. Field and petrologic data presented here demonstrate that models of subduction initiation based on the IBM forearc are currently simplistic. We advocate that geodynamic models of subduction initiation along Philippine Sea Plate margins incorporate a pre-Eocene, north-verging subduction zone (the proto West Philippine Trench of Faccenna et al., 2010) associated with Cretaceous terranes forming the overriding plate, a doubly-vergent subduction initiation as well as the interplay of plate forces and mantle upwelling (e.g. Oki-Daito plume of Ishizuka et al. 2013) during incipient subduction.

## 6 Conclusions

1. The extrusive sequence in the juvenile arc section of Zambales ophiolite in Luzon Island, Philippines consists of boninite and boninite-series volcanics. This makes northern Zambales ophiolite the lone middle Eocene boninite locality in the Philippine Sea Plate apart from the Ogasawara Islands, Guam and their submarine equivalent in the IBM forearc.

2. Zambales boninite and boninite series volcanics are marked by low trace element abundances relative to mid-ocean ridge basalt with hydrous fluid mobile element enrichment and heavy rare earth element depletion comparable with boninite from Troodos and Oman. Trace element ratios suggest that aqueous fluids released from shallow sub-arc depths are the dominant slab components in Zambales boninite.

3. Subduction initiation stratigraphy deduced from diving, dredging and drilling programs in the Izu-Ogasawara (Bonin)-Mariana forearc is recognized in Zambales ophiolite as well. The temporal variation from proto-arc basalt to boninite and post-boninite arc tholeiite is represented by the Coto (45 Ma), Barlo (44 Ma) and Sual-Subic (44–43 Ma) volcanic sections of Zambales ophiolite. Incipient subduction associated with the juvenile arc in northern Zambales ophiolite likely involved the unfolded East Asian Sea crust as subducting slab and the westernmost margin of Philippine Sea Plate as the overriding plate.

4. By studying Zambales ophiolite, it can be shown that subduction initiation (SI) is a plate-scale process. To address subduction initiation, it is essential to consider not just the Izu-Ogasawara-Mariana forearc but geologic constraints from the entire Philippine Sea Plate as well. Doubly-vergent subduction initiation deduced from the rock record is a testable mechanism that can be explored by geodynamic modelling.





## Data availability

The data associated with this publication are available in the Supplement.

Supplementary Table 1: Representative microprobe analyses of olivine, pyroxene and spinel in Zambales boninite.

Supplementary Table 2: Whole rock compositions of boninite and boninite-series volcanics from northern Zambales
ophiolite.

## Competing interests

The authors declare that they have no conflict of interest.

## Acknowledgements

This work forms part of the first author's research project under a Ministry of Education, Culture, Sports, Science and
Technology (MEXT) PhD scholarship. Field expedition to Zambales ophiolite was supported by a 2016 Fukada Geological
Institute Grant-in Aid to A. Perez. This work was also supported by JSPS KAKENHI Grant no. 25400446 and donated funds
from JAPEX to S. Umino, and no. 25287133 to O. Ishizuka. We thank F.T. Jumawan, A. Mapa, B.D. Payot, J.M.R. Guotana
and E.G. Gadot for their assistance during the March 2015 reconnaissance and May 2016 field campaigns and K. Yamanobe
for her help during geochemical analyses at GSJ.

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

**Figure 1: (a) Distribution of subduction initiation rock suites in the Philippine Sea Plate and spreading histories of associated marginal basins (Deschamps and Lallemand, 2002; Pearce et al., 1992; Reagan et al., 2017). (b) Regional geological map of Zambales ophiolite after Yumul et al., (1990) with whole rock K-Ar and zircon U-Pb ages from Fuller et al., (1989) and**
15 **Encarnacion et al., (1993). The 44−43 Ma Acoje Block consists of the northern Masinloc massif and the southernmost San Antonio massif. Enclosed region is the Barlo extrusive section mapped in this study.**



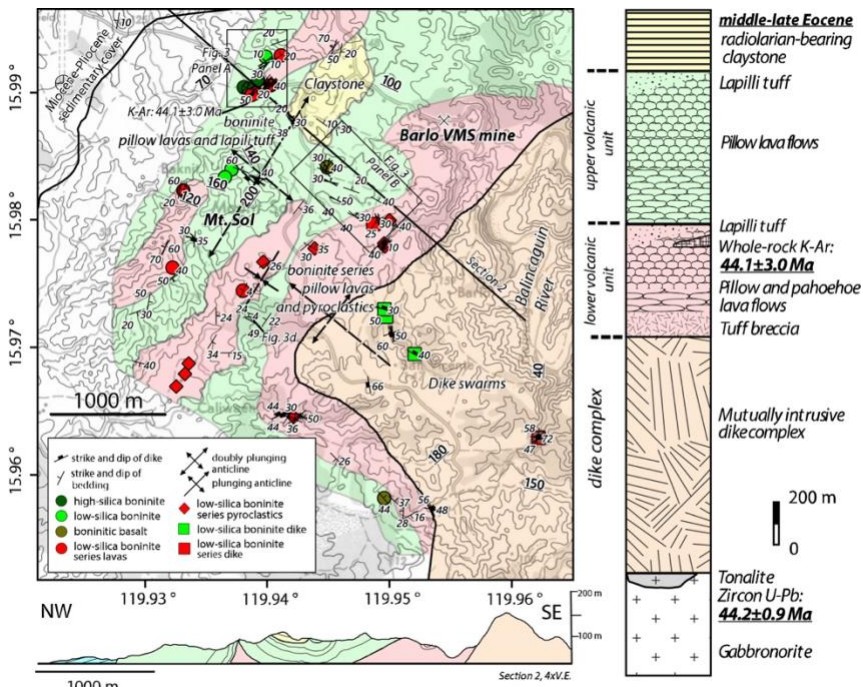

**Figure 2: Geological map of Barlo and environs. Closed symbols are samples with whole-rock geochemical data. Enclosed areas (panel A and B) are shown in Figure 3. Geologic cross-section is given along a NW-SE section line. Geologic column shows stratigraphic relationships and estimated thickness of the upper and lower volcanic unit.**



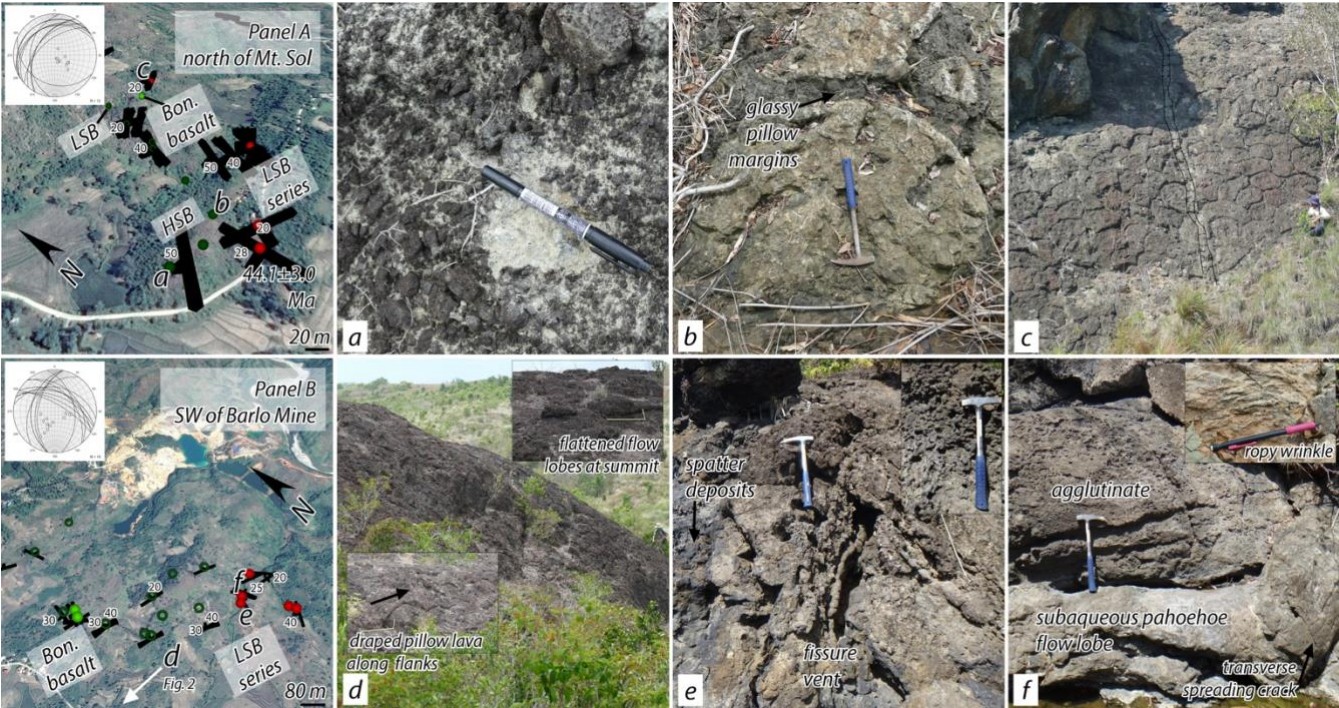

Figure 3: Representative outcrops showing distinct volcanic facies and stratigraphic relationship of the upper boninite unit and lower basaltic andesite-andesite unit. (a) High-Si boninite lapilli tuff. (b) High-Si boninite pillow lava with fresh glassy margins. (c) Basaltic andesite pillow lavas cut by boninite dike. (d) Boninite pillow volcano. (e) Fissure vent with fluidal, flattened basaltic andesite-andesite spatter deposits and lapilli tuff. (f) Basaltic andesite pahoehoe flow with ropy wrinkles (inset) and transverse spreading cracks on lobe crust indicating flow direction. Rock hammer for scale (0.42 m). Orientation poles of bedding planes (dip azimuth) are shown in lower hemisphere stereographic projection (Vollmer, 2015). Bedding plane orientations in GoogleEarth (Digital Globe, CNES/Astrium) images, viewed from southeast direction, are visualized using the S2K macro (Blenkinsop, 2012).



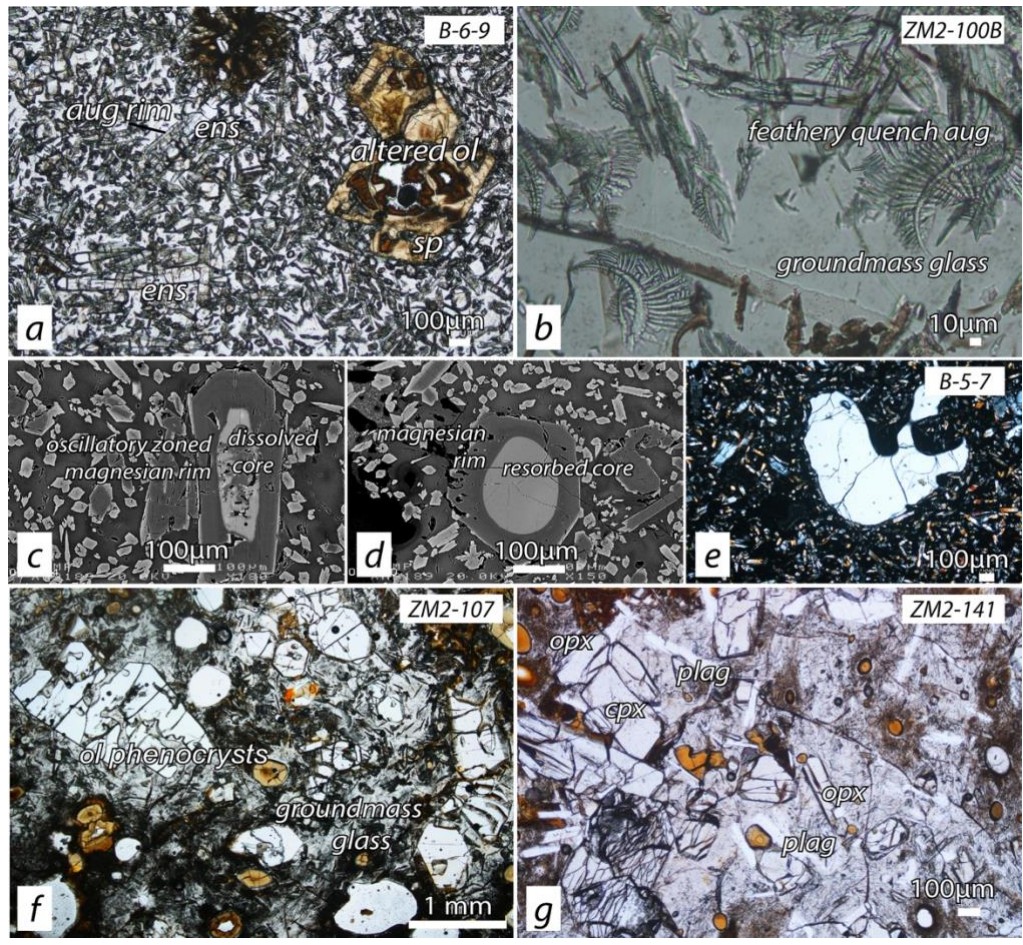

**Figure 4: Microphotographs showing textures of Zambales boninite and boninite-series volcanics. (a) High-silica boninite (HSB) with altered olivine and abundant elongate enstatite microphenocryts with augite rims (plane polarized-light). (b) Characteristic feathery clinopyroxene in groundmass glass, HSB (plane polarized-light). (c) Oscillatory zoning in magnesian rim of enstatite xenocryst with dissolved core, HSB (BSE). (d) Reverse zoning in enstatite xenocryst with resorbed core, HSB (BSE). (e) Embayed quartz xenocryst, HSB (crossed polars). (f) Low-silica boninite (LSB) with fresh olivine phenocrysts in glassy groundmass (plane polarized-light). (g) Plagioclase, othropyroxene and clinopyroxene microphenocrysts in LSB-series basaltic andesite tuff breccia matrix (plane polarized-light).**





**Figure 5: Comparative mineral chemistry of constituent phases (olivine, orthopyroxene, clinopyroxene and spinel) in Zambales and Ogasawara boninites. Mantle olivine array is from Takahashi, (1986).**



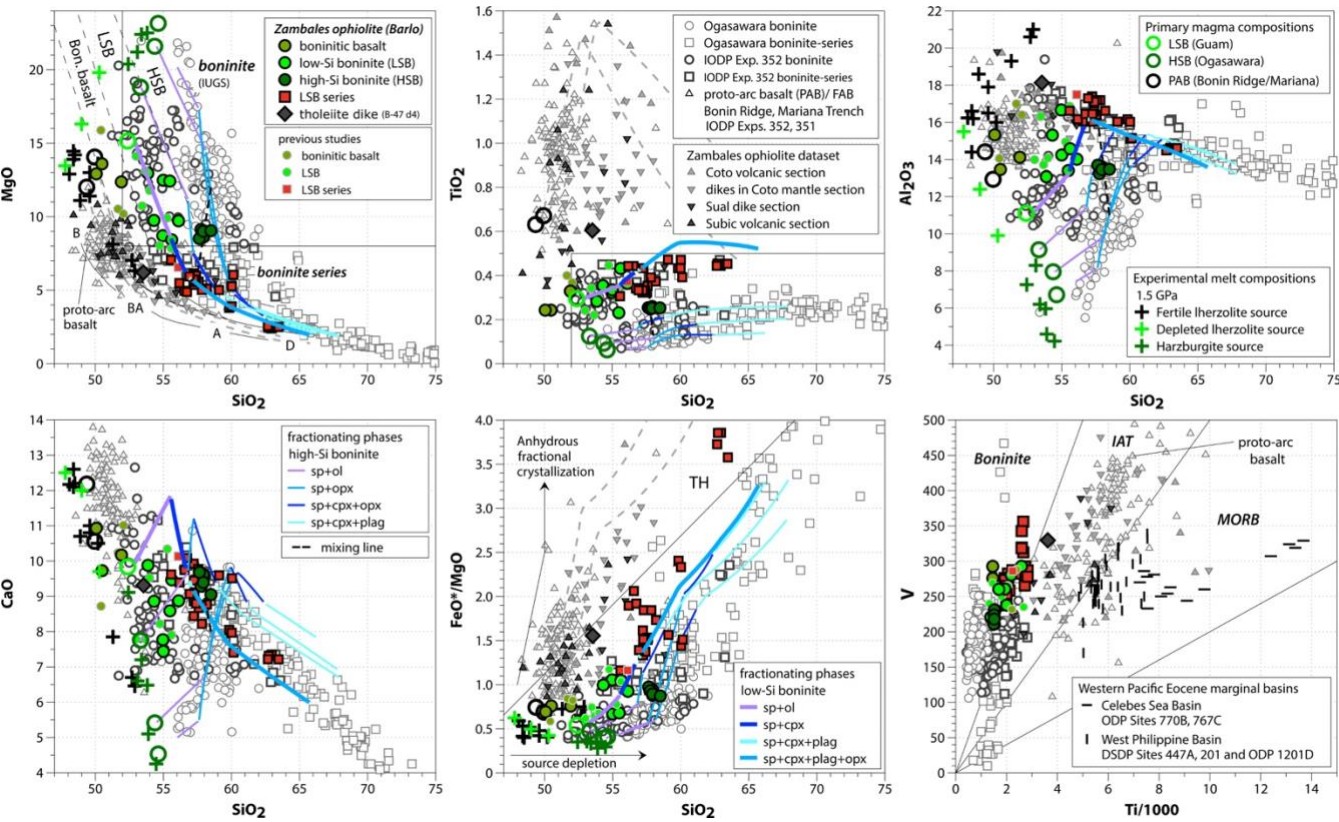

**Figure 6: MgO, TiO₂, Al₂O₃, CaO, FeO\*/MgO vs. SiO₂ variation diagrams showing the distinct parental magma compositions and modelled fractional crystallization paths of high- and low-silica boninites and boninite-series volcanics. Fractional crystallization paths are modelled at +1 Δlog(FMQ), 1 kb, 1 wt% H₂O, 1500-980 °C. Modified boninite classification after Kanayama et al. (2013) and Reagan et al. (2015). Dividing lines between boninite and basalt-andesite-dacite-rhyolite (BADR) series are from Pearce and Robinson (2010). Ti-V systematics (Shervais, 1982) of IBM and Zambales subduction initiation rock suites is also shown.**





**Figure 7: Extended MORB and chondrite normalized trace element patterns. Normalizing values are from Barrat et al. (2012) and Gale et al. (2013).**





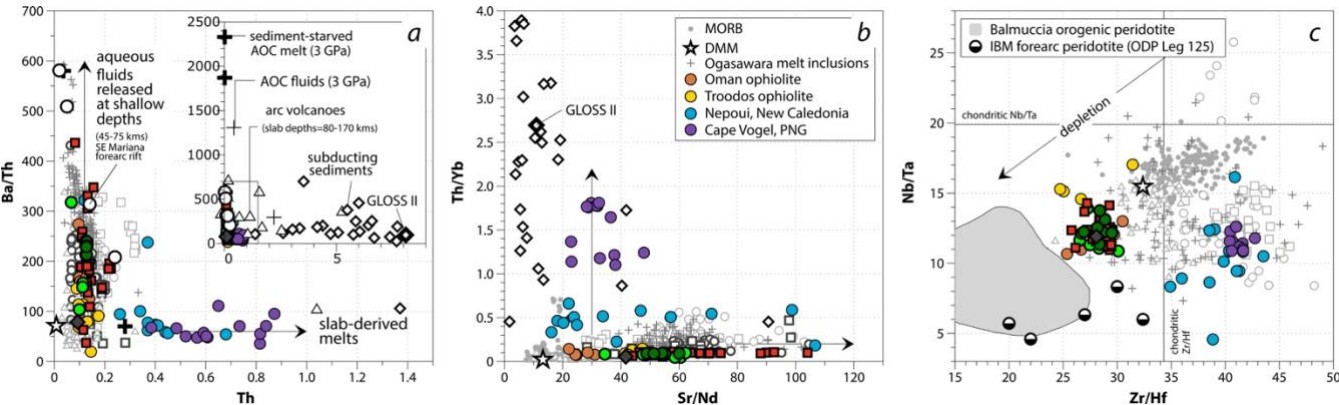

**Figure 8: Trace element ratios discriminating the nature of slab-derived components in boninite and boninite-series volcanics. Zambales, Oman and Troodos boninites have distinct LILE and HFSE enrichments compared to Ogasawara, Cape Vogel and Nepoui. Data sources: Ogasawara spinel-hosted boninitic melt inclusions (Umino et al., 2017), altered oceanic crust (AOC) fluids, melts and sediments (Carter et al., 2015), aqueous fluids released at shallow depths (Ribeiro et al., 2015), subducting sediments and GLOSS II (Plank, 2014). Chondritic bulk Earth Nb/Ta and Zr/Hf ratios are from Munker et al., (2003). DMM and MORB compositions from Jenner and O'Neill, (2012) and Workman and Hart, (2005), respectively. Symbols are the same as in Fig. 6 except those shown in the diagram.**

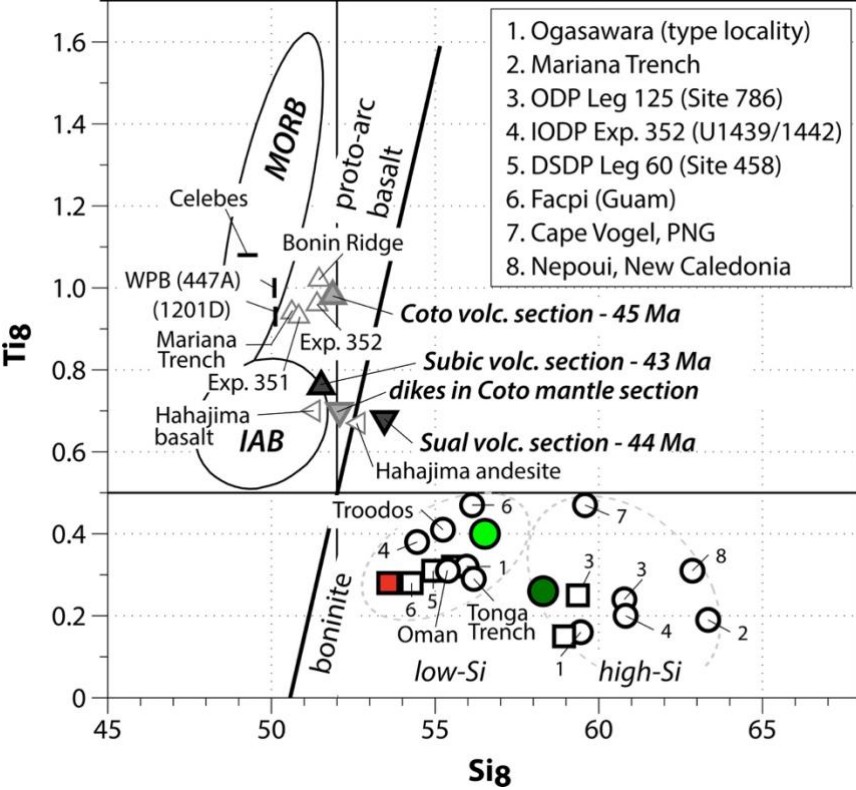

**Figure 9: 8wt% MgO-normalized $SiO_2$ ($Si_8$) and $TiO_2$ ($Ti_8$) values of the crustal sections of Zambales ophiolite and IBM forearc showing temporal evolution from proto-arc basalt, boninite to arc tholeiite associated with subduction initiation. Also shown are $Si_8$-$Ti_8$ values of other notable boninite localities (Oman, Troodos, Tonga Trench). MORB and IAB fields are from Pearce and Robinson (2010). Data sources are given in the text. Symbols are the same as in Fig. 6 except those shown in the diagram.**



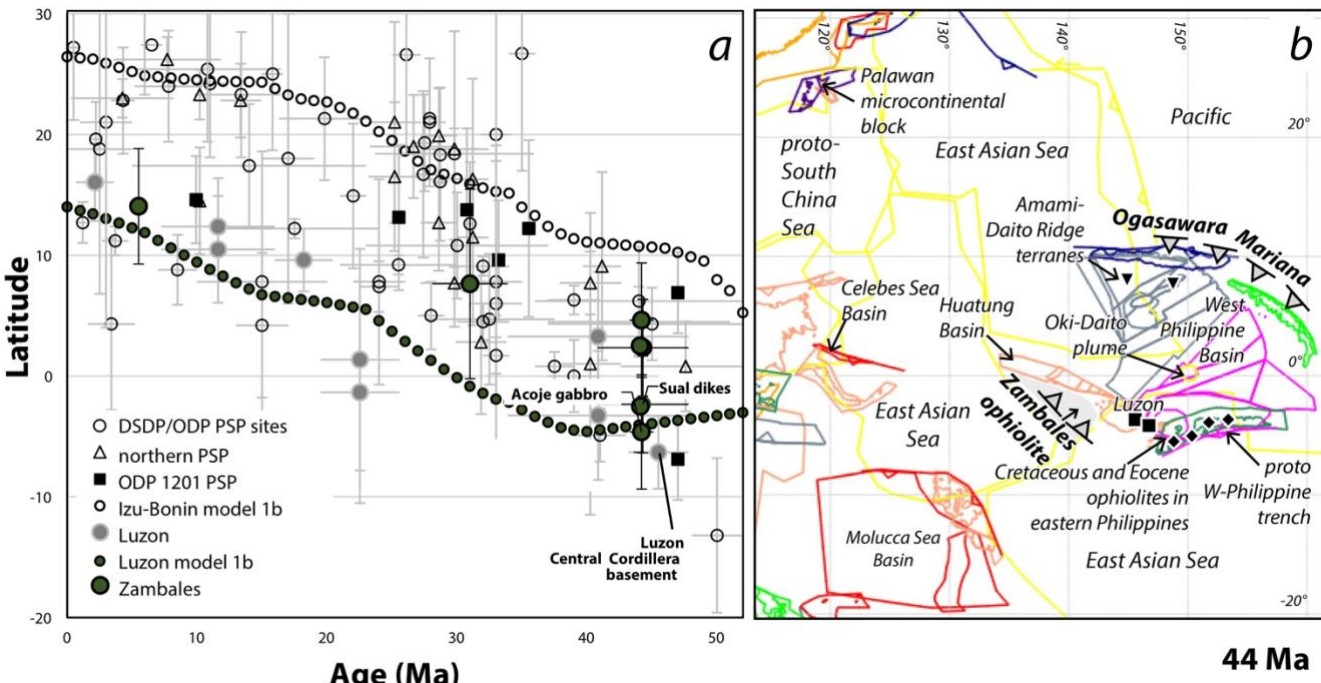

**Figure 10: (a) Paleolatitudes derived from Philippine Sea Plate, Luzon and Zambales inclination data (Fuller et al., 1991; Hall et al., 1995; Queano et al., 2007; Queano et al., 2009; Richter and Ali, 2015; Yamazaki et al., 2010) (b) Tectonic reconstruction of the western Pacific region at 44 Ma using GPlates (Boyden et al., 2011) and unfolded slab data, plate polygons and rotation file (model 1b) of Wu et al. (2016).**