# Peer review of "Boninite and boninite-series volcanics in northern Zambales ophiolite: Doubly-vergent subduction initiation along Philippine Sea Plate margins"

_Solid Earth, 2017_

## Referee Comment (RC1) · S. Whattam (Referee) · 20 Mar 2018

Scott A. Whattam Whattam Geosciences 335 Roxdale Avenue Orleans, Ontario K1E 1T9 Canada e-mail: sawhatta@gmail.com phone: 1-613-818-9389

Review of Solid Earth Discussions Manuscript: Boninite and boninite-series volcanics in northern Zambale: Doubly-vergent subduction initiation along Philippine Sea Margins, by Perez, A. et al. Reviewed by Scott A. Whattam: Submitted: 20-March-2018

c/o Editor Dear Editor,

[Figure]

I have now finished my review of the aforementioned SED Manuscript which I provide below.

MS DESCRIPTION & SYNOPIS On the basis of new field, geochemical, and mineral chemistry constraints, the manuscript explains the relevance of newly discovered high-Si boninites comprising the Zambales ophiolite as it pertains to subduction initiation scenarios for the region.

BROADER IMPACT The ms will be of great interest to those working on the tectonic evolution of the western Pacific region as it provides key chemostratigraphic evidence to link NE-dipping subduction initiation (SI) to SW dipping SI at the IBM forearc circa 50-45 Ma. PRESENTATION & SCIENTIFIC INTERPRETATION Overall the text is well-written and free of grammatical errors; suggestions for improvement of the text are provided in minor instances and listed under "specific points" below. One point is that "the" needs to be used more often but it would be a daunting task to indicate every-where in the ms this is required. Most of the figures are of high-quality; an exception is Fig. 6 which way "too busy"; there is too much data plotted on the figure. Would suggest plotting perhaps only data from this study. As well, there are some symbols in the plots which are not identified in the legend. For example in the first panel (MgO vs. SiO2, these different plots should be labelled a, b, c) the solid triangles are not described.

The main problem I have with the ms concerns the treatment of the tectonic configuration at and after subduction initiation (SI) and the concept of the "doubly-vergent" SI. This (doubly-vergent SI) is mentioned in the title and addressed in the final section of the manuscript. However, the data provided in this ms (essentially geochemical) cannot address this. I think the authors should probably just drop this section altogether or consider/provide alternative models for SI. An alternative title could be: Boninite and boninite-series volcanics in the northern Zambales ophiolite: Implications for subduction initation along Philippine Sea Plate margins. For example, there is no mention of a possible plume-induced SI scenario yet Figure 10a shows the Oki-Daito Plume smack

in the middle (beneath) the WPB at (and probably just before?) subduction initiation. I believe that Wu et al. posit that the WPB formed as the result of plume emplacement pretty much at the same time or just before SI. An explanation for the cause of doubly-vergent subduction is not provided in the ms; I find the similar timing of SI on either sides of the WPB very difficult to explain without a plume-induced origin. Evidence for this (plume-contaimnation and hence a possible plume-induced SI scenario) would be in the form of isotopes from the proto-arc basalts and boninites which should record plume-contamination if there was a plume-induced origin (I am pretty certain that IBM FAB do not record evidence of plume-contamination). In any event, the SI scenario at the WPB appears to be similar to that of the Late Cretaceous along the Caribbean Large Igneous Province (CLIP). Whattam and Stern (2015) suggested that SI was likely plume-induced and resulted in subduction along a great portion of the periphery of the CLIP. The difference for this ms however, was that evidence for plume contamination was shown. Even if the authors do not address the doubly-vergent SI, the ms is still of great value as it documents a chemostratigraphy/chemotemporality identical to that of the IBM forearc.

SPECIFIC POINTS 1. Page 1, Abstract: Line 15: as this is the first discovery of hi-Si boninite in the Zambales ophiolite, this should be explicitly stated. 2. Page 1, Abstract: Line 18: place "the" before "Zambales ophiolite"; this has to be done in many instances throughout the ms 3. Page 1, towards bottom of the abstract: Perhaps should state that work on the Coto Block was done by others and not by this study; I had to subsequently look through the ms to see if work was done on both the Coto and Acoje blocks 4. Page 1, Line 31: would insert "and vice versa" at end of first sentence 5. Page 2, Line 2: after "plume-induced subduction initiation" should reference Whattam and Stern (2015) and Gerya et al. (2016) (I believe Whattam and Stern (2015) were the first to specifically coin this) 3 6. Page 2, Line 6: Don't understand what "Challenge No. 11" means 7. Page 2, Line 7: Would replace "including" with "with the exception of" 8. Page 2, Line 23: replace "verified" with "suggested"; we believe this to be the case, yes, but can't outright verify it 9. Introduction focuses almost exclusively of subduction

initiation (SI) at the IBM; relevant, but SI has also been discussed elsewhere; as well, different ideas of how SI transpires → e.g., spontaneous vs. induced (Stern, 2004) 10. Page 3, Line 27: don't understand what "transitional" MORB means; transitional to MORB and IAT? If so, state this. As well, the IBM FAB which may be analogous to Coto Block MORB-like lavas, have characteristics intermediate to and which overlap MORB and IAT (e.g., Whattam et al., submitted). For example, whole-rock chemistry documents a an arc-like Ti-V ratio <20 and evidence of melting of an source more oxidized than MORB (higher Fe3+/FeT, Brounce et al., 2015). More on relation between Coto Block lavas and FAB later. 11. Page 4, Line 7: sentence ending with "transition zone" needs references. 12. Page 4, Line 24: I believe boninitic basalts was also mentioned earlier? These need to be defined at first instance (i.e., lavas which record MgO >8 wt. % and TiO2 <0.5 wt. % as per IUGS boninite definition but SiO2 <52 wt. %) 13. Page 5, Line 17: Again, confused as to whether paper included Coto Block; Maybe state in first sentence of this paragraph that study was conducted on Acoje Block (only) 14. Page 5, Line 24: change "lost weight" → weight lost 15. Page 5, Line 31: Spell out GSJ/AIST 16. Page 6: Section 5.1: This section is very "dense" and difficult to read. I suggest making a table showing the different lithologies and their mineralogy and textures, and then significantly shortening the written description here 17. Page 6, Line 15: Should of probably brought this up earlier, but similar to point 12 above, perhaps all the different categories of boninite (low-Si, high-Si, basaltic etc.) should be explained in the introduction or at least at an earlier point in ms 18. Page 6, Line 20: I think this is the first mention of Ogasawara? Mention where this is- Japan → part of IBM forearc? 19. Page 6, Line 24: change is → are 20. Page 7 Line 6: Insert "A" before peculiar 4 21. Page 7, Line 15: These LOI are very high. And what rock types exhibit these values? All or just high-Si boninites? These should be in the Table in Supp. Doc., correct? So put Supp. Doc. X at end of sentence. Maybe a plot of LOI versus various potentially mobile elements (e.g., MgO, K2O, Na2O, Ba) is warranted? Or at least some sort of statement like "though the LOI are high our petrologic arguments are based primarily on trace elements known to be immobile up

to greenschist-facies conditions". Are any filters being applied to your samples? For example, using only samples which yield 98-102 wt.% oxides or ones with <3 % LOI? 22. Page 7, Line 18: What is primary? And I note here that primary is used later on but not defined. Do you mean primary lavas such that exhibit high MgO, high Mg# ($\sim\geq65$), high Cr and high Ni? 23. Page 7, Line 22: I think a reference is needed after "boninitic basalts". Maybe Pearce and Robinson, or Reagan et al. (2017)? Not sure. 24. Page 7, Line 23: As mentioned at the beginning, Figure 6 is very "busy". Would suggest plotting only samples from this study. 25. Page 7, Line 24: change second "and" → or 26. Page 7, Line 29: pristine? You mention above LOI values of 4-7 wt.%. 27. Page 7, Line 31: ug/g? usually reported in ppm 28. Page 8, Line 3: replace within with → between 29. Page 8, Line 15: Change so reads: Compositions of Zambales boninite.....are marked by low incompatible trace element abundances... 30. Page 8, Line 23: replace times with → x 31. Page 8, Line 25: insert boninite between Zambales and ophiolite 32. Page 8, Line 32: descending order is unclear; perhaps describe from base → top which is probably standard convention 33. Page 9, Line 6: Is unclear how can be classified as moderate-Fe tholeiites without the Miyahsiro plot overlain by Arculus' low-med-high Fe series fields 34. Page 9, Line 20: Haugen (2017): Is this a MSc or PhD thesis (not indicated in references). 35. Page 10, Line 4: I think a paragraph at least is warranted to explain how the modelling was done using MELTS (supplementary document probably appropriate). 36. Page 10, Line 5: Ghiorso and Gualda (2015) not in references 37. Page 10, Line 16: change "in the base" to → at the base 5 38. Page 10, Line 17: break sentence; add ";" after "at depth" 39. Page 10, Line 20: change slightly deviate → deviate slightly 40. Page 10, Line 24: change does → do 41. Page 10, bottom of page: Would change Section 6.2 title to → Slab contributions 42. Page 10, bottom of page: Would include more up-to-date references for boninite petrogenesis 43. Page 11, Line 7-8: OK, but they are equally LREE-depleted. What is the explanation for the spoon-shaped REE patterns? I think for the "classical" U-shaped signatures that the explanation is high-degree partial melting (which produces low MREE) which is subsequently slab-fluid modified to

produce LREE enrichment; not sure of explanation for the high HREE 44. Page 11, Line 13: Maybe explain at beginning of section that Ba/Th is a marker/gauge of shallow slab-contributions and reference (Pearce et al., ?) Why Ba liberated at shallow conditions? Low temperature (I think); low P as well? 45. Page 11, Line 14: insert "increasing" after "mirrored by" 46. Page 11, Line 15: reference Fig. 8b after Th/Yb 47. Page 11, Line 16: what is decoupled? 48. Page 11, Line 18: Insert "A" before "high U/Th ratio" 49. Page 11, Line 19: change ratio → ratios; change by → if; add "to source" at end of sentence 50. Page 11, Line 21: the La/Th vs. Sm/La is not shown so have to indicate this 51. Page 11, Lines 30-31: Why mention slab melts? This is not mentioned previously and I don't think anyone familiar with boninite petrogenesis would consider slab melts as part of the equation. 52. Page 11, Line 2: change so reads: transitional between MORB and IAT 53. Page 12, Line 3: Change "in" → on the basis of 54. Page 12, Lines 3-4: Confusing sentence; why mention distinct from Mariana BAB? 55. Page 12, Line 5: Another confusing sentence; have to get point across that depletion in REEs, TiO2, Zr and Y of Acoje relative to Coto documents the progressive depletion of...what about LILE enrichments? These should increase from Coto → Acoje 56. Page 12, Line 10: Can't readily see where Coto lavas plot in Ti/V space but this is a very important point as FAB can be distinguished by MORB on the basis of Ti/V which is arc-like (>20) and by virtue of elevated Fe3+/FeT indicative of a more oxidized (arc-like) source. Suggest you state what the Ti/V 6 ratios of the Coto lavas are and compare these with those of IBM. Are they similar or not? 57. Page 12, Section 6.4. See the Presentation and Scientific Interpretations section 58. Page 13, Line 12: insert "above a west-dipping subduction zone" after (Ishizuka et al., 2011) 59. Page 13, Lines 15-20: sounds perhaps like a plume-induced SI scenario 60. Page 14, Lines 6-8: Why feasible? No explanation for this (doubly-vergent subduction) 61. Page 14, Line 11: Change north-verging → NE-verging

Please also note the supplement to this comment:
https://www.solid-earth-discuss.net/se-2017-138/se-2017-138-RC1-supplement.pdf

---

## Referee Comment (RC2) · PhD Encarnacion (Referee) · 28 Apr 2018

Review of "Boninite and boninite-series volcanics in the northern Zambales ophiolite: Doubly-vergent subduction initiation along Philippine Sea Plate margins" by Perez et al. for Solid Earth Discussion

This paper provides new structural, stratigraphic, petrographic, and geochemical data from the northern Zambales ophiolite, Philippines, that is an important contribution to our understanding of this important terrane in the western Pacific. The resulting

geochemical/petrological stratigraphy is compared with the Izu-Bonin-Mariana forearc crust and a tectonic model is proposed to explain the origin of the Zambales ophiolite.

The paper has two main parts. The first part is a very well-documented detailed study of the volcanic facies preserved in the ophiolite crust combined with detailed geochemical analyses of the crustal rocks. This section is well-written and does not need much revision; I have just minor corrections, questions, and suggestions listed below. The second part is the tectonic model, which, in my opinion has issues that need to be addressed, especially the relationship of the Zambales with the rest of Luzon as discussed below (page 13, lines 21-35). Although I personally do not agree with the proposed model, I think it will invite debate and stimulate further study that will advance our understanding of this important area in the western Pacific.

I recommend publication with moderate revision focusing on the tectonic model.

Page 1 Line 13: Delete "uniquely" and just say "predominant."

Page 2 Lines 2-4: The statement that Cenozoic subduction initiation (SI) requires pre-existing weak zones or lithospheric collapse is erroneous. All models of SI require a weak zone, even lithospheric collapse. Lines 9-10: I wouldn't say that the IBM is "the most appropriate locality" to test models of subduction initiation. It might be the most appropriate place to test the Stern and Bloomer-type model, but there are other models as well. Maybe say "one of the most appropriate. . ." Lines 24-25: Cite references for "spontaneous" (e.g., Leng and Gurnis) and "induced" (e.g., Hall et al.) SI that have been replicated by numerical modeling. Line 28: delete semi-colon Line 30: Maybe change "unequivocally be attributed" to "uniquely explain."

Page 3 Line 1: In discussing the basement complexes of the Philippines, Encarnacion (2004) might be an appropriate paper to cite. (Encarnacion, 2004, Multiple ophiolite generation preserved in the northern Philippines. Tectonophysics.) Line 17: Change "structurally bound" to "fault-bound" (if that is what is meant by "structurally bound"). Lines 21-22: In reference to "terrane docking," I presume the Zambales ophiolite is the

terrane. But what is it docking with? (Indeed, what do you mean by "docking"?) Line 23: Change "in the westernmost margin" to "just west" of the ophiolite. The Mesozoic cherts are not found in the ophiolite itself, but in mélange-type shear zone material west of the ophiolite. Line 23-25: Besides Queano et al. 2017, you should also cite Hawkins and Evans (1983) who first described these cherts and Encarnacion (2004) who provides additional description of the shear zone. Line 24: The shear zone is not "buried;" it is well-exposed in several places.

Page 4 Lines 2-4: You might want to note that the idea that the San Antonio massif is a displaced block from the north is disputed by Encarnacion et al. (1999) for lack of convincing evidence. Line 18: You may want to add that the arc affinity of the Acoje block is also consistent with radiogenic isotope data (Pb, Sr, and Nd), which indicate hydrous fluid enrichment (Encarnacion et al., 1999). Line 24: What rock types were the "bedding planes" measured on? Line 26: Change to "...NW-plunging anticline just south." Line 28: What does "conjugate intrusive directions" mean? (Intruded into conjugate fractures sets?). Please clarify. Line 32: Change to "subaqeous fall-out deposits" (?)

Page 5 Lines 3-4: Do the terms "Strombolian to Hawaiian fire fountaining" apply to subaqeous/submarine volcanics? I would think not(?) Are these deposits in fact sub-aerial? If so, that would be a very important finding! It would imply that portions of the Zamabales ophiolite in fact contain what might be actual subaerial island arc crust, which would substantially change the paleogeography of Luzon. Please clarify. Line 12: What are the "upright structures"? Sedimentary/volcaniclastic bedding? Please clarify.

Page 6 Line 1: Change "12" to "Twelve." (Avoid numerals at the beginning of a sentence.) Line 2: Add "Standard JB-2 and JB-3 . . ."

Page 7 Line 15: Change to "The Cs, Rb, . . . and Mn contents in . . ."

Page 8 Line 16: Change to "...MORB for most trace elements..."

Page 10 Line 32: Please add a few words summarizing the general consensus on boninite genesis. ("...there is general consensus on boninite petrogenesis, namely that..."

Page 12 Line 3: Change semi-colon to a period. Line 7: Change "Ma" to "Myr." (You are referring to differences in ages, not a point in time.)

Page 13 Lines 2-5: Regarding the paleomagnetic data, do the declinations support a Philippine Sea Plate connection as well? The Philippine Sea Plate has been in-ferred to have rotated clockwise. There are additional previous paleomagnetic studies (for example, Fuller et al., 1991, J. Asian Earth Sciences) that appear to demonstrate counterclockwise rotation for Luzon. Line 20: By "rapid emplacement" do you mean "rapid formation"? If not, what do you mean by "rapid emplacement"? Line 20: Change to "The timing of the proposed subduction inititation..." Lines 21-35 to lines 1-3 of page 14: This section discusses the relationship between the Zambales ophiolite and Eocene (Angat) and Cretaceous ophiolites in the east Luzon area. The authors state that the similarity in ages of the Zambales and Angat ophiolites presented in Encar-nacion et al. (1993) "does not necessarily prove and affinity" between the two. But Encarnacion et al. (1993) (and Encarnacion et al., 1999, and Encarnacion, 2004) did not use the ages alone in arguing that the Zambales and Angat ophiolites (and Creta-ceous ophiolites) are contiguous (not exotic/allochthonous to each other). The geology and stratigraphy as well are consistent with the Zambales-Angat ophiolite forming adja-cent to the Cretaceous ophiolite in the east, which was the main point of Encarnacion et al., 1993 (and reiterated/amplified in Encarnacion, 2004). Regarding the issues mentioned above, I think suggesting that the Zambales is exotic to east Luzon causes greater problems with the proposed model, because the east Luzon ophiolites and arc crust are in-between the Zambales and the Philippines Sea Plate. In other words, if one wants to separate the Zambales ophiolite from east Luzon, shouldn't it also be separate from the Philippine Sea Plate?

Page 14 Lines 5-7: Why is doubly-vergent subduction "feasible"? Please elaborate.

Line 29: Change to "By studying the Zambales ophiolite. . ." Line 29: Delete "(SI)" Line 29: It is stated that subduction initiation is a "plate-scale process." I'm not sure what the purpose of this statement is. When is it not a plate-scale process? Please clarify.

Figure 2: The caption should include the references for the ages shown in the figure.

Figure 2: Page 5 says the 44.1 Ma age is from a sill. This isn't clear or indicated in the stratigraphic column.

Figure 10: In panel "b", what are the diamonds, inverted triangles, and squares?

— end —

Please also note the supplement to this comment:
https://www.solid-earth-discuss.net/se-2017-138/se-2017-138-RC2-supplement.pdf

---

## Author Comment (AC1) · 7 May 2018

**Authors' reply to RC #1**

We would like to thank the reviewer for the insightful comments.

In the first half of this response we address the main comments raised by Referee 1 (Dr. Scott Whattam).

Referee comments are in black and our responses are in blue.

The main problem I have with the ms concerns the treatment of the tectonic configuration at and after subduction initiation (SI) and the concept of the "doubly-vergent" SI. This (doubly-vergent SI) is mentioned in the title and addressed in the final section of the manuscript. However, the data provided in this ms (essentially geochemical) cannot address this. I think the authors should probably just drop this section altogether or consider/provide alternative models for SI. An alternative title could be: Boninite and boninite-series volcanics in the northern Zambales ophiolite: Implications for subduction initiation along Philippine Sea Plate margins.

We use "doubly-vergent SI along Philippine Sea Plate margins" in a descriptive way, synonymous to "subduction initiation on both sides of the PSP". If doubly-vergent SI will be sustained it will lead to "double in-dip subduction" (Holt et al., 2017).

We disagree with the reviewer that our data cannot address "doubly-vergent SI". As detailed in our MS; the petrological, geochemical and most importantly field characterization of boninite, and the recognition of subduction initiation stratigraphy in the Zambales ophiolite when considered with available published geological and geophysical data (Section 6.4) support the initiation of subduction on both sides of the PSP.

Subduction along the margins of the Philippine Sea Plate is already well established (Deschamps and Lallemand, 2002, 2003; Lallemand, 2016) and a NE-verging subduction zone on the western margin of the PSP is reflected in both global and regional plate reconstructions of SE Asia and the western Pacific region (Wu et al., 2016; Zahirovic et al., 2014, 2016)

What we provide in our MS is evidence from the rock record for the inception of such a subduction zone.

We use and highlight "doubly-vergent SI" in the title because this configuration is distinct from SI scenarios solely based on the IBM forearc which mainly focuses on the problem of whether subduction initiation is spontaneous or induced (e.g. Arculus et al., 2016; Keenan and Encarnación, 2016).

Geodynamic models of SI should be based on robust geologic data and a "doublyvergent SI" configuration based on the Zambales ophiolite provides another "boundary condition" for refined models of SI along PSP margins.

The alternative would be placing the Zambales ophiolite along the IBM forearc, similar to Fig. 4 of Pearce et al., (1992). From geologic arguments alone, (i.e. the opening of the West Philippine Basin at 55-46 Ma and presence of the Philippine archipelago in between) this would be untenable.

For example, there is no mention of a possible plume-induced SI scenario yet Figure 10a shows the Oki-Daito Plume smack in the middle (beneath) the WPB at (and probably just before?) subduction initiation. I believe that Wu et al. posit that the WPB formed as the result of plume emplacement pretty much at the same time or just before SI. An explanation for the cause of doubly-vergent subduction is not provided in the ms; I find the similar timing of SI on either sides of the WPB very difficult to explain without a plume-induced origin. Evidence for this (plumecontaimnation and hence a possible plume-induced SI scenario) would be in the form of isotopes from the proto-arc basalts and boninites which should record plumecontamination if there was a plume-induced origin (I am pretty certain that IBM FAB do not record evidence of plume-contamination). In any event, the SI scenario at the WPB appears to be similar to that of the Late Cretaceous along the Caribbean Large Igneous Province (CLIP). Whattam and Stern (2015) suggested that SI was likely plume-induced and resulted in subduction along a great portion of the periphery of the CLIP. The difference for this ms however, was that evidence for plume contamination was shown. Even if the authors do not address the doubly-vergent SI, the ms is still of great value as it documents a chemostratigraphy/chemotemporality identical to that of the IBM forearc.

What our data cannot directly address is the association of the "doubly-vergent SI" configuration along PSP margins to "plume-induced subduction initiation/ PISI". Thus, it is not mentioned in the MS and we make no attempt to link the two.

The location of Oki-Daito plume (based on Wu et al., 2016) is indicated in Fig. 10b for the only reason that in Page 14 Line 14-15 (original MS) we suggest that "the interplay between plate forces and mantle upwelling (e.g. Oki-Daito plume of Ishizuka et al., 2013) should be explored by geodynamic models".

We agree with the reviewer that a "doubly-vergent SI" configuration might provide a link to "plume-induced subduction initiation". However, at this point, to explicitly associate the two would be highly speculative and it is more appropriate to address this problem in a separate paper. As an example, the geodynamic models of Baes et al., (2016) can be modified to specifically test/demonstrate PISI in the PSP. The size and spreading history of the West Philippine Basin and the crustal structure of Cretaceous and Eocene terranes are constrained well enough that these can be incorporated in the models. The model dimensions can be set-up as 2640 km x 1525 km x 2640 km, and the crustal thickness of the overriding Cretaceous arc terranes-Eocene intra-arc basins as 15-25 km and 10-15 km, respectively (Nishizawa et al., 2014; Wu et al., 2016). But in our opinion and as demonstrated by Baes et al., (2016) the most critical parameter will be the size and shape of the Oki-Daito plume which is, so far, unknown. The age difference of the overriding and subducting plates might also be a key.

Using PRIMELT3 (Herzberg and Asimow, 2015), the primary magma composition and olivine liquidus temperature of a depleted (proto-arc) basalt (PAB) from the Coto Block that precedes boninite from the Acoje Block can likewise be estimated. The olivine liquidus temperature of a depleted PAB from Coto Block lies in an adiabatic upwelling path similar to Mariana Trench PAB with a Tp of ~1370 °C (Perez et al., in prep). Ambient (MORB) and plume (Mauna Loa) mantle potential temperatures are estimated by Putirka, (2016) as 1330-1450 °C and 1560-1670 °C, respectively. So far, we find no evidence for the excess heat, thermal anomaly and plume head size invoked by Macpherson and Hall, (2001); hence our reticence to discuss PISI.

Isotopic investigations are in agreement that a mantle plume is likely associated with the oceanic plateaus and bathymetric highs emplaced at the same time as the opening of the West Philippine Basin (Hickey-Vargas et al., 2006.; Ishizuka et al., 2013). However, these studies also have ruled out this plume as the source of excess heat for boninite generation and concluded that a link between the Oki-Daito mantle plume and subduction initiation is rather unlikely. It is not the case that we didn't show any evidence of plume-induced SI but it is a question of whether there is one, at least from a petrological point of view.

Our main point is that characterization of the Eocene SI along Philippine Sea Plate margins as "doubly-vergent" is still valid whether it is plume-induced or not.

In our view, the most prudent way to address this is to note that the geometry of incipient subduction with convergent subduction zones on both sides/ along its margins as deduced from the Eocene tectonic reconstruction of the Philippine Sea Plate resembles the results of geodynamic models of plume-induced subduction initiation (Baes et al., 2016; Gerya et al., 2015). And we have added new lines in the revised manuscript to reflect this.

" Although current terrestrial subduction is dominantly asymmetric, it is interesting to note that two-sided subduction is what is essentially produced in 2-D and 3-D models of mantle convection (Gerya et al., 2008; Wada & King, 2015). Likewise, we discern that doubly-vergent SI geometry or subduction initiation with oppositely-dipping subduction zones along its margins is characteristic of 3-D thermo-mechanical models of plume-induced subduction initiation (Baes et al., 2016; Gerya et al., 2015). Plumeinduced subduction initiation was first recognized by Whattam and Stern, (2015) to describe the temporal association of late Cretaceous plume-related oceanic plateaus in Central America that was followed by arc volcanism with oppositely dipping slab dispositions. In the case of the Eocene Western Pacific, a mantle plume centered on the Manus Basin had originally been invoked by Macpherson and Hall, (2001) to account for an inferred thermal anomaly in IBM boninite mantle source, implicitly connecting the initiation of the IBM arc to the presence of a mantle plume. Isotopic studies are in agreement that a mantle plume is likely associated with the oceanic plateaus and bathymetric highs emplaced at the same time as the opening of the West Philippine Basin (Hickey-Vargas et al., 2006; Ishizuka et al., 2013); however, thermal anomalies in excess of the ambient mantle are not reflected in mantle potential temperature estimates of proto-arc basalt from the IBM forearc (Umino et al., 2017). A causative link between a mantle plume and the doubly-vergent SI configuration along Philippine Sea Plate margins is yet to be established. We speculate that the location of Philippine Sea Plate (PSP) in the nexus of Pacific, Indo-Australian and Eurasian plates and their long-term Cenozoic plate motion makes doubly-vergent subduction initiation along its margins feasible. The northwestward translation and clockwise rotation of the Philippine Sea Plate starting in the early Eocene had to be accommodated by the adjoining oceanic domain east of southern Eurasia (e.g. East Asian Sea); hence, its interaction with the oceanic leading edge of the Philippine Sea Plate is expected (Wu et al., 2016; Zahirovic et al., 2016) and likely led to incipient subduction (Fig. 10). The dynamics of sustained double-vergent subduction is examined by Holt et al. (2017) but doubly-vergent subduction initiation is yet to be explored by numerical modelling. Field and petrologic data presented here demonstrate that models of subduction initiation based on the IBM forearc are currently simplistic. Geodynamic models of subduction initiation should be based on robust geologic data and a doubly-vergent SI configuration based on the Zambales ophiolite provides another boundary condition for refined models of SI along PSP margins. We advocate that geodynamic models of subduction initiation along Philippine Sea Plate margins incorporate a pre-Eocene, N/NE-dipping subduction zone (the proto West Philippine Trench of Faccenna et al., 2010) associated with Cretaceous terranes forming the overriding plate, doubly-vergent subduction initiation along its margins as well as the interplay of plate forces and mantle upwelling (e.g. Oki-Daito plume of Ishizuka et al. 2013) during incipient subduction (Fig. 10b).is a testable mechanism that can be explored by geodynamic modelling."

In the next section, we address the points given by the reviewer in the "Specific points" section. We've adopted most of the suggestions given by the reviewer.

SPECIFIC POINTS

1. Page 1, Abstract: Line 15: as this is the first discovery of hi-Si boninite in the Zambales ophiolite, this should be explicitly stated. Sentence modified  $\rightarrow$  "We report for the first time..."

2. Page 1, Abstract: Line 18: place "the" before "Zambales ophiolite"; this has to be done in many instances throughout the ms We reviewed the manuscript and placed "the" in places we've missed.

3. Page 1, towards bottom of the abstract: Perhaps should state that work on the Coto Block was done by others and not by this study; I had to subsequently look through the ms to see if work was done on both the Coto and Acoje blocks Reference added

4. Page 1, Line 31: would insert "and vice versa" at end of first sentence "and vice versa" added

5. Page 2, Line 2: after "plume-induced subduction initiation" should reference Whattam and Stern (2015) and Gerya et al. (2016) (I believe Whattam and Stern (2015) were the first to specifically coin this) Added Whattam and Stern (2015). Gerya et al. (2016) is probably Gerya et al. (2015)? which was cited previously

6. Page 2, Line 6: Don't understand what "Challenge No. 11" means As explicitly stated in the IODP (International Ocean Discovery Program) Science Plan for 2013-2013 "Challenge No. 11" is "How do subduction zones initiate, cycles volatiles, and generate continental crust?"

7. Page 2, Line 7: Would replace "including" with "with the exception of"

replaced with "with the exception of"

8. Page 2, Line 23: replace "verified" with "suggested"; we believe this to be the case, yes, but can't outright verify it replaced "verified" with "suggested"

9. Introduction focuses almost exclusively of subduction initiation (SI) at the IBM; relevant, but SI has also been discussed elsewhere; as well, different ideas of how SI transpires  $\rightarrow$  e.g., spontaneous vs. induced (Stern, 2004) This is briefly discussed in Page 2 Lines 24-31

10. Page 3, Line 27: don't understand what "transitional" MORB means; transitional to MORB and IAT? If so, state this. As well, the IBM FAB which may be analogous to Coto Block MORB-like lavas, have characteristics intermediate to and which overlap MORB and IAT (e.g., Whattam et al., submitted). For example, whole-rock chemistry documents an arc-like Ti-V ratio <20 and evidence of melting of a source more oxidized than MORB (higher Fe3+/FeT, Brounce et al., 2015). More on relation between Coto Block lavas and FAB later. Sentence modified

Addressed fully in #56

11. Page 4, Line 7: sentence ending with "transition zone" needs references. References added

12. Page 4, Line 24: I believe boninitic basalts was also mentioned earlier? These need to be defined at first instance (i.e., lavas which record MgO >8 wt. % and TiO2 <0.5 wt. % as per IUGS boninite definition but SiO2 <52 wt. %) Sentence modified

13. Page 5, Line 17: Again, confused as to whether paper included Coto Block; Maybe state in first sentence of this paragraph that study was conducted on Acoje Block (only)

Sentence changed to "For this study, a subset of forty-four (44) samples located along NW-SE transects of the Acoje Block volcanic sequence at Barlo were selected for whole rock geochemical analyses and screened through visual and microscopic assessment of secondary alteration."

14. Page 5, Line 24: change "lost weight"  $\rightarrow$  weight lost changed to "weight lost"

15. Page 5, Line 31: Spell out GSJ/AIST

now "Geological Survey of Japan/National Institute of Advanced Industrial Science and Technology"

16. Page 6: Section 5.1: This section is very "dense" and difficult to read. I suggest making a table showing the different lithologies and their mineralogy and textures, and then significantly shortening the written description here Seems ok to us

17. Page 6, Line 15: Should of probably brought this up earlier, but similar to

point 12 above, perhaps all the different categories of boninite (low-Si, high-Si, basaltic etc.) should be explained in the introduction or at least at an earlier point in ms

18. Page 6, Line 20: I think this is the first mention of Ogasawara? Mention where this is- Japan  $\rightarrow$  part of IBM forearc?

Sentence changed to "This assemblage corresponds to Type II boninite of Umino (1986) in samples described from the type locality at Chichijima island, Ogasawara (Bonin) Archipelago in southern Japan."

19. Page 6, Line 24: change is  $\rightarrow$  are changed

20. Page 7 Line 6: Insert "A" before peculiar changed

21. Page 7, Line 15: These LOI are very high. And what rock types exhibit these values? All or just high-Si boninites? These should be in the Table in Supp. Doc., correct? So put Supp. Doc. X at end of sentence. Maybe a plot of LOI versus various potentially mobile elements (e.g., MgO, K2O, Na2O, Ba) is warranted? Or at least some sort of statement like "though the LOI are high our petrologic arguments are based primarily on trace elements known to be immobile up to greenschist-facies conditions". Are any filters being applied to your samples? For example, using only samples which yield 98-102 wt.% oxides or ones with <3 % LOI?

Yes, indeed the LOI values are a bit high. We note that in general boninites have LOI values that are greater than 3 wt%.

IODP Expedition 352 shipboard boninite samples have LOI values that are much higher (mostly between 5-11 wt%) and samples from Haugen, 2017 which are the "cleanest, least weathered pieces" of the shipboard cores have LOI values that are just as much as the samples from this study.

Concerning element mobility, this is addressed in Page 7 lines 15-19.

Filters have already been applied in the current dataset. The current dataset was screened not only based on LOI values but primarily based on element immobility judged using diagrams such as CaO vs. Na2O, major oxide vs LOI, and major oxide/trace element vs. immobile element (e.g. Zr). Samples with mobile element enrichment (e.g. total alkalis, Rb) followed a vertical vector and were excluded.

22. Page 7, Line 18: What is primary? And I note here that primary is used later on but not defined. Do you mean primary lavas such that exhibit high MgO, high Mg# ( $\cong$ 65), high Cr and high Ni? Sentence modified.

23. Page 7, Line 22: I think a reference is needed after "boninitic basalts". Maybe Pearce and Robinson, or Reagan et al. (2017)? Not sure. Reagan et al. (2015) added

24. Page 7, Line 23: As mentioned at the beginning, Figure 6 is very "busy". Would suggest plotting only samples from this study.

We slightly modified Figure 6. We removed objects that are not discussed in the text such the primary PAB and PAB fractionation. However, we've chosen to still include

the Ogasawara and Expedition 352 boninite and boninite series datasets. Our point is to compare Zambales boninite with the dataset from the IBM forerac. Another point is to show the major element variation vis-à-vis the modelled fractional crystallization path. We believe these are better addressed if our data are shown with the Ogasawara and Expedition 352 datasets. The panels are now marked as "a, b, c"

---

## Author Comment (AC2) · 7 May 2018

**Authors' reply to RC #2**

We would like to thank the reviewer for the insightful comments.

In this response, we address the main comments raised by Referee 2 (Dr. John Encarnacion).

Referee comments are in black and our responses are in blue.

Page 1 Line 13: Delete "uniquely" and just say "predominant." We feel that it is indeed unique, especially the occurrence of high-silica boninite. This point is raised by Reagan et al., (2017) as well.

Page 2 Lines 2-4: The statement that Cenozoic subduction initiation (SI) requires pre-existing weak zones or lithospheric collapse is erroneous. All models of SI require a weak zone, even lithospheric collapse.

This statement is a general characterization to differentiate spontaneous and induced SI from plume-induced SI.

Page 2 Lines 9-10: I wouldn't say that the IBM is "the most appropriate locality" to test models of subduction initiation. It might be the most appropriate place to test the Stern and Bloomer-type model, but there are other models as well. Maybe say "one of the most appropriate..."

Sentence modified  $\rightarrow$  "...one of the most appropriate..."

Page 2 Lines 24-25: Cite references for "spontaneous" (e.g., Leng and Gurnis) and "induced" (e.g., Hall et al.) SI that have been replicated by numerical modeling. These references were cited in the following sentences (original MS page 2 lines 25-29.

Page 2 Line 28: delete semi-colon Semi-colon deleted.

Page 2 Line 30: Maybe change "unequivocally be attributed" to "uniquely explain." Sentence modified.

Page 3 Line 1: In discussing the basement complexes of the Philippines, Encarnacion (2004) might be an appropriate paper to cite. (Encarnacion, 2004, Multiple ophiolite generation preserved in the northern Philippines. Tectonophysics.) Additional references added.

Page 3 Line 17: Change "structurally bound" to "fault-bound" (if that is what is meant by "structurally bound"). Changed to "fault-bound"

Page 3 Lines 21-22: In reference to "terrane docking," I presume the Zambales ophiolite is the terrane. But what is it docking with? (Indeed, what do you mean by "docking"?)

Here terrane docking is used in a way similar to Pubellier et al., (2004). We mean

that by the Pliocene its transport from equatorial positions to its present location probably have ceased. The identity of what Zambales ophiolite "docked with" is still unknown. It may be a composite of the following (1) an ocean basin that predates the South China Sea (pre-2016 there is consensus that this is proto-South China Sea), the vestiges of which are the Jurassic-Cretaceous cherts in the West Luzon Shear Zone of Karig, (1983) and complemented by recent radiolarian studies of Queaño et al., (2017a, 2017b) and (2) the northern extension of the Palawan microcontinental block (based on seismic data, Arfai et al., (2011) speculates that West Luzon Basin west of Zambales ophiolite is partly underlain by continental basement).

Page 3 Line 23: Change "in the westernmost margin" to "just west" of the ophiolite. The Mesozoic cherts are not found in the ophiolite itself, but in mélange-type shear zone material west of the ophiolite.

The limited distribution of these cherts maybe more aptly described with "in the westernmost margin" rather than "just west" of the ophiolite.

Page 3 Line 23-25: Besides Queano et al. 2017, you should also cite Hawkins and Evans (1983) who first described these cherts and Encarnacion (2004) who provides additional description of the shear zone.

Additional references added - Hawkins and Evans (1983), Karig et al., (1986) and Encarnacion (2004).

Page 3 Line 24: The shear zone is not "buried;" it is well-exposed in several places. Sentence modified

Page 4 Lines 2-4: You might want to note that the idea that the San Antonio massif is a displaced block from the north is disputed by Encarnacion et al. (1999) for lack of convincing evidence.

I re-visited the Encarnacion et al., (1999) paper but I can't seem to find the lines that dispute idea that the San Antonio massif is a displaced block from the north. In Encarnacion (2004) page 119, it is stated that "Except for some possible tectonic displacements of locally derived blocks (Yumul et al, 1998) the ophiolite is a coherent slab...".

Page 4 Line 18: You may want to add that the arc affinity of the Acoje block is also consistent with radiogenic isotope data (Pb, Sr, and Nd), which indicate hydrous fluid enrichment (Encarnacion et al., 1999).

This reference is cited in the discussion section 6.2

Page 4 Line 24: What rock types were the "bedding planes" measured on? Bedding planes were measured primarily on pillow lavas and tuff breccias.

Page 4 Line 26: Change to "...NW-plunging anticline just south." Seems ok to us.

Page 4 Line 28: What does "conjugate intrusive directions" mean? (Intruded into conjugate fractures sets?). Please clarify. (Dikes with) conjugate intrusive directions pertain to obliquely intruding dikes.

Line 32: Change to "subaqeous fall-out deposits" (?) Changed.

Page 5 Lines 3-4: Do the terms "Strombolian to Hawaiian fire fountaining" apply to subaqeous/submarine volcanics? I would think not(?) Are these deposits in fact subaerial? If so, that would be a very important finding! It would imply that portions of the Zamabales ophiolite in fact contain what might be actual subaerial island arc crust, which would substantially change the paleogeography of Luzon. Please clarify. Here we are referring to submarine Strombolian to Hawaiian-type fire fountaining. Sentence is modified and a reference discussing deep submarine pyroclastic eruptions (Head and Wilson, 2003) is added.

Page 5 Line 12: What are the "upright structures"? Sedimentary/volcaniclastic bedding? Please clarify.

"Upright structures" include flattened flow lobes in the summit of the recognized pillow volcano shown in Fig3d (inset).

"Upright" is used in page 5 line 12 to describe the primary igneous structures (emplacement-related) in the synclinal area in a broad sense. It is used in contrast to post-ophiolite emplacement folding affecting the volcanic section and overlying sedimentary sequences.

Page 6 Line 1: Change "12" to "Twelve." (Avoid numerals at the beginning of a sentence.)

Sentence modified.

Page 6 Line 2: Add "Standard JB-2 and JB-3 ..." Sentence modified.

Page 7 Line 15: Change to "The Cs, Rb, ... and Mn contents in ..." Sentence modified.

Page 8 Line 16: Change to "...MORB for most trace elements..." I think this is not necessary.

Page 10 Line 32: Please add a few words summarizing the general consensus on boninite genesis. ("...there is general consensus on boninite petrogenesis, namely that..." Sentence modified.

Page 12 Line 3: Change semi-colon to a period.

Sentence modified.

Line 7: Change "Ma" to "Myr." (You are referring to differences in ages, not a point in time.) Sentence modified.

Page 13 Lines 2-5: Regarding the paleomagnetic data, do the declinations support a Philippine Sea Plate connection as well? The Philippine Sea Plate has been inferred to have rotated clockwise. There are additional previous paleomagnetic studies (for

example, Fuller et al., 1991, J. Asian Earth Sciences) that appear to demonstrate counterclockwise rotation for Luzon.

Declination-based interpretations of the Cenozoic rotation history of Luzon and Zambales are contentious; Fuller et al., (1991) argues for CCW rotation while McCabe et al., (1987) calls for CW rotation.

As discussed by the Queano et al., (2007), the use of declination data in tectonic models of Luzon and Zambales might be of limited use since the CW and CCW directions cannot be unambiguously ascribed to either local or major plate rotations.

Page 13 Line 20: By "rapid emplacement" do you mean "rapid formation"? If not, what do you mean by "rapid emplacement"?

We do not mean rapid formation. Rapid emplacement is inferred based on the reasoning that juvenile arc magmatism progressed only up to boninite; that it is a case of "failed' subduction initiation or arrested arc development. On the other hand, the IBM forearc which can be described as a case of "successful" subduction initiation that produced an intra-oceanic arc with calc-alkaline lavas (41-35 Ma) after boninite (48-44 Ma) and proto-arc basalt (52-48 Ma).

Page 13 Line 20: Change to "The timing of the proposed subduction inititation..." Sentence modified.

Page 13 Lines 21-35 to lines 1-3 of page 14: This section discusses the relationship between the Zambales ophiolite and Eocene (Angat) and Cretaceous ophiolites in the east Luzon area. The authors state that the similarity in ages of the Zambales and Angat ophiolites presented in Encarnacion et al. (1993) "does not necessarily prove and affinity" between the two. But Encarnacion et al. (1993) (and Encarnacion et al., 1999, and Encarnacion, 2004) did not use the ages alone in arguing that the Zambales and Angat ophiolites (and Cretaceous ophiolites) are contiguous (not exotic/allochthonous to each other). The geology and stratigraphy as well are consistent with the Zambales-Angat ophiolite forming adjacent to the Cretaceous ophiolite in the east, which was the main point of Encarnacion et al., 1993 (and reiterated/amplified in Encarnacion, 2004).

We disagree with the interpretations of Encarnacion et al. (1993).

In particular, we invite the reviewer to place the west dipping Cretaceous-Eocene subduction zone east of Sierra Madre that produced a late Cretaceous arc and an Eocene arc-backarc basin pair (presumably the Zambales ophiolite) shown in Fig.12 of Encarnación, (2004) in current global and regional plate reconstructions of SE Asia and the western Pacific region (Wu et al., 2016; Zahirovic et al., 2014, 2016).

Indeed, age constraints were not the sole parameter used in arguing that the Zambales and Angat ophiolites (and Cretaceous ophiolites) are contiguous. But the Eocene volcaniclastic sequence, as shown in Fig.6 of Encarnación, (2004), overlies the Angat (Eocene) and Montalban (Cretaceous) ophiolites and not the Zambales ophiolite. Bachman et al., (1983) notes that there is marked lithological difference between the east and west flanks of the Luzon Central Valley Basin (CVB); he further characterized the CVB as a forearc basin.

Regarding the issues mentioned above, I think suggesting that the Zambales is exotic to east Luzon causes greater problems with the proposed model, because the east Luzon ophiolites and arc crust are in-between the Zambales and the Philippines Sea Plate. In other words, if one wants to separate the Zambales ophiolite from east Luzon, shouldn't it also be separate from the Philippine Sea Plate?

On the contrary, we find no problem if east Luzon ophiolites and arc crust are placed in-between the Zambales ophiolite and the Philippine Sea Plate. With regards to Zambales ophiolite being "separated" from the Philippines Sea Plate, this is the case in earlier tectonic reconstructions which follows the arc- backarc basin scenario where Zambales ophiolite is alternatively placed east of the Celebes Sea Basin in contact with the western margin of the Philippine Sea Plate (Fig.5 of Rangin et al., 1990a and Plate 2 of Rangin et al., 1990b).

In its present form, the G-Plates based reconstruction in Fig 10b using the unfolded slab data, plate polygons and rotation file of Wu et al. (2016) is actually compatible with Encarnacion (2004) in a broad sense. If Zambales ophiolite was formed by the proposed incipient subduction on the western margin of the Philippine Sea Plate, one can argue that Zambales ophiolite can be contiguous with basement of Luzon and that it is relatively autochthonous (with respect to Luzon) (Encarnación, 2004); albeit highly displaced (relative to its present position) caused by transport along a plate boundary (Pubellier et al., 2004). A back-arc origin for Eocene ophiolites associated with Cretaceous arc in eastern Philippines is compatible with Fig. 10b as well.

Arguments against a backarc basin origin for Zambales are given in section 6.3; the only contention would then be the relationship of Angat to Zambales ophiolite.

Page 14 Lines 5-7: Why is doubly-vergent subduction "feasible"? Please elaborate.

We speculate that the location of Philippine Sea Plate (PSP) in the nexus of Pacific, Indo-Australian and Eurasian plates and their long-term Cenozoic plate motion makes doubly-vergent subduction initiation along its margins feasible. The northwestward translation and clockwise rotation of the Philippine Sea Plate starting in the early Eocene had to be accommodated by the adjoining oceanic domain east of southern Eurasia (e.g. East Asian Sea); hence, its interaction with the oceanic leading edge of the Philippine Sea Plate is expected (Wu et al., 2016; Zahirovic et al., 2016) and likely led to incipient subduction (Fig. 10).

Page 14 Line 29: Change to "By studying the Zambales ophiolite..." Sentence modified.

Page 14 Line 29: Delete "(SI)" Deleted.

Page 14 Line 29: It is stated that subduction initiation is a "plate-scale process." I'm not sure what the purpose of this statement is. When is it not a plate-scale process? Please clarify.

The purpose of the statement is to emphasize that subduction initiation may not be localized in the eastern margin of the Philippine Sea Plate. The doubly-vergent SI configuration presented here is distinct from current SI scenarios solely based on the

IBM forearc which mainly focuses on the problem of whether subduction initiation is spontaneous or induced (e.g. Arculus et al., 2016; Keenan and Encarnación, 2016).

Figure 2: The caption should include the references for the ages shown in the figure. References added.

Figure 2: Page 5 says the 44.1 Ma age is from a sill. This isn't clear or indicated in the stratigraphic column.

It is shown in the stratigraphic column.

Figure 10: In panel "b", what are the diamonds, inverted triangles, and squares? Captions modified.

**References**

Arculus, R. J., Ishizuka, O., Bogus, K. A., Gurnis, M., Hickey-Vargas, R., Aljahdali, M. H., Bandini-Maeder, A. N., Barth, A. P., Brandl, P. A., Drab, L., do Monte Guerra, R., Hamada, M., Jiang, F., Kanayama, K., Kender, S., Kusano, Y., Li, H., Loudin, L. C., Maffione, M., Marsaglia, K. M., McCarthy, A., Meffre, S., Morris, A., Neuhaus, M., Savov, I. P., Sena, C., Tepley III, F. J., van der Land, C., Yogodzinski, G. M., Zhang, Z., Keenan, T. E., Encarnacion, J., Arculus, R. J., Ishizuka, O., Bogus, K. A., Gurnis, M., Hickey-Vargas, R., Aljahdali, M. H., Bandini-Maeder, A. N., Barth, A. P., Brandl, P. A., Drab, L., do Monte Guerra, R., Hamada, M., Jiang, F., Kanayama, K., Kender, S., Kusano, Y., Li, H., Loudin, L. C., Maffione, M., Marsaglia, K. M., McCarthy, A., Meffre, S., Morris, A., Neuhaus, M., Savov, I. P., Sena, C., Tepley III, F. J., van der Land, C., Yogodzinski, G. M., Zhang, Z., Keenan, T. E. and Encarnacion, J.: Unclear causes for subduction, Nat. Geosci., 9(5), 338 [online] Available from: http://dx.doi.org/10.1038/ngeo2703, 2016.

Arfai, J., Franke, D., Gaedicke, C., Lutz, R., Schnabel, M., Ladage, S., Berglar, K., Aurelio, M., Montano, J. and Pellejera, N.: Geological evolution of the West Luzon Basin (South China Sea, Philippines), Mar. Geophys. Res., 32(3), 349–362, doi:10.1007/s11001-010-9113-x, 2011.

Bachman, S. B., Lewis, S. D. and Schweller, W. J.: Evolution of a forearc basin, Luzon Central Valley, Philippines, Am. Assoc. Pet. Geol. Bull., 67(7), 1143–1162, 1983.

Encarnación, J.: Multiple ophiolite generation preserved in the northern Philippines and the growth of an island arc complex, Tectonophysics, 392(1–4), 103–130, doi:10.1016/J.TECTO.2004.04.010, 2004.

Fuller, M., Haston, R., Lin, J. L., Richter, B., Schmidtke, E. and Almasco, J.: Tertiary paleomagnetism of regions around the South China Sea, J. Southeast Asian Earth Sci., 6(3–4), 161–184, doi:10.1016/0743-9547(91)90065-6, 1991.

Head, J. W. and Wilson, L.: Deep submarine pyroclastic eruptions: theory and predicted landforms and deposits, J. Volcanol. Geotherm. Res., 121(3–4), 155–193, doi:10.1016/S0377-0273(02)00425-0, 2003.

Karig, D. E.: Accreted terranes in the northern part of the Philippine archipelago, Tectonics, 2(2), 211–236, doi:10.1029/TC002i002p00211, 1983.

Keenan, T. E. and Encarnación, J.: Unclear causes for subduction, Nat. Geosci., 9(5), 338–338, doi:10.1038/ngeo2703, 2016.

McCabe, R., Kikawa, E., Cole, J. T., Malicse, A. J., Baldauf, P. E., Yumul, J. and

Almasco, J.: Paleomagnetic results from Luzon and the central Philippines, J. Geophys. Res., 92(B1), 555, doi:10.1029/JB092iB01p00555, 1987.

Pubellier, M., Monnier, C., Maury, R. and Tamayo, R.: Plate kinematics, origin and tectonic emplacement of supra-subduction ophiolites in SE Asia, Tectonophysics, 392(1–4), 9–36, doi:10.1016/J.TECTO.2004.04.028, 2004.

Queano, K. L., Ali, J. R., Milsom, J., Airchison, J. C. and Pubellier, M.: North Luzon and the Philippine Sea Plate motion model: Insights following paleomagnetic, structural, and age-dating investigations, J. Geophys. Res., 112, 2007.

Queaño, K. L., Marquez, E. J., Dimalanta, C. B., Aitchison, J. C., Ali, J. R. and Yumul, G. P.: Mesozoic radiolarian faunas from the northwest Ilocos Region, Luzon, Philippines and their tectonic significance, Isl. Arc, 26(4), e12195–n/a, doi:10.1111/iar.12195, 2017a.

Queaño, K. L., Dimalanta, C. B., Yumul, G. P., Marquez, E. J., Faustino-Eslava, D. V, Suzuki, S. and Ishida, K.: Stratigraphic units overlying the Zambales Ophiolite Complex (ZOC) in Luzon, (Philippines): Tectonostratigraphic significance and regional implications, J. Asian Earth Sci., 142, 20–31,

doi:10.1016/j.jseaes.2016.06.011, 2017b.

Rangin, C., Jolivet, L. and Pubellier, M.: A simple model for the tectonic evolution of Southeast Asia and Indonesia region for the past 43 m.y, Bull. la Soc. Geol. Fr., VI(6), 889–905, doi:10.2113/gssgfbull.VI.6.889, 1990a.

Rangin, C., Pubellier, M., Azema, J., Briais, A., Chotin, P., Fontaine, H., Huchon, P., Jolivet, L., Maury, R., Muller, C., Rampnoux, J. P., Stephan, J. F., Tournon, J., Cottereau, N., Dercourt, J. and Ricou, L. E.: The quest for Tethys in the western Pacific; 8 paleogeodynamic maps for Cenozoic time, Bull. la Soc. Geol. Fr., VI(6), 907–913, doi:10.2113/gssgfbull.VI.6.907, 1990b.

Reagan, M. K., Pearce, J. A., Petronotis, K., Almeev, R. R., Avery, A. J., Carvallo, C., Chapman, T., Christeson, G. L., Ferré, E. C., Godard, M., Heaton, D. E.,

Kirchenbaur, M., Kurz, W., Kutterolf, S., Li, H., Li, Y., Michibayashi, K., Morgan, S., Nelson, W. R., Prytulak, J., Python, M., Robertson, A. H. F., Ryan, J. G., Sager, W. W., Sakuyama, T., Shervais, J. W., Shimizu, K. and Whattam, S. A.: Subduction initiation and ophiolite crust: new insights from IODP drilling, Int. Geol. Rev., 59(11), 1439–1450, doi:10.1080/00206814.2016.1276482, 2017.

Wu, J., Suppe, J., Lu, R. and Kanda, R.: Philippine Sea and East Asian plate tectonics since 52 Ma constrained by new subducted slab reconstruction methods, J. Geophys. Res. Solid Earth, 121(6), 4670–4741, doi:10.1002/2016JB012923, 2016. Zahirovic, S., Seton, M. and Müller, R. D.: The Cretaceous and Cenozoic tectonic evolution of Southeast Asia, Solid Earth, 5(1), 227–273, doi:10.5194/se-5-227-2014, 2014.

Zahirovic, S., Matthews, K. J., Flament, N., Müller, R. D., Hill, K. C., Seton, M. and Gurnis, M.: Tectonic evolution and deep mantle structure of the eastern Tethys since the latest Jurassic, Earth-Science Rev., 162, 293–337, doi:10.1016/J.EARSCIREV.2016.09.005, 2016.

---

## Author Response (AR1)

**Authors' reply to RC #1**

SPECIFIC POINTS

1. Page 1, Abstract: Line 15: as this is the first discovery of hi-Si boninite in the Zambales ophiolite, this should be explicitly stated.

5 Sentence modified  $\rightarrow$  "We report for the first time..."

2. Page 1, Abstract: Line 18: place "the" before "Zambales ophiolite"; this has to be done in many instances throughout the ms

We reviewed the manuscript and placed "the" in places we've missed.

3. Page 1, towards bottom of the abstract: Perhaps should state that work on the Coto Block

10 was done by others and not by this study; I had to subsequently look through the ms to see if work was done on both the Coto and Acoje blocks

Reference added

4. Page 1, Line 31: would insert "and vice versa" at end of first sentence "and vice versa" added

15 5. Page 2, Line 2: after "plume-induced subduction initiation" should reference Whattam and Stern (2015) and Gerya et al. (2016) (I believe Whattam and Stern (2015) were the first to specifically coin this)

Added Whattam and Stern (2015). Gerya et al. (2016) is probably Gerya et al. (2015)? which was cited previously

20 6. Page 2, Line 6: Don't understand what "Challenge No. 11" means

As explicitly stated in the IODP (International Ocean Discovery Program) Science Plan for 2013-2013 "Challenge No. 11" is "How do subduction zones initiate, cycles volatiles, and generate continental crust?"

7. Page 2, Line 7: Would replace "including" with "with the exception of"

**25 replaced with "with the exception of"**

8. Page 2, Line 23: replace "verified" with "suggested"; we believe this to be the case, yes, but can't outright verify it

1

replaced "verified" with "suggested"

9. Introduction focuses almost exclusively of subduction initiation (SI) at the IBM; relevant, but SI has also been discussed elsewhere; as well, different ideas of how SI transpires  $\rightarrow$  e.g., spontaneous vs. induced (Stern, 2004)

This is briefly discussed in Page 2 Lines 24-31

5 10. Page 3, Line 27: don't understand what "transitional" MORB means; transitional\_to MORB and IAT? If so, state this. As well, the IBM FAB which may be analogous to\_Coto Block MORB-like lavas, have characteristics intermediate to and which overlap\_MORB and IAT (e.g., Whattam et al., submitted). For example, whole-rock chemistry\_documents an arc-like Ti-V ratio <20 and evidence of melting of a source more\_oxidized than MORB (higher Fe3+/FeT, 10 Brounce et al., 2015). More on relation</li>

between Coto Block lavas and FAB later.

Sentence modified

Addressed fully in #56

11. Page 4, Line 7: sentence ending with "transition zone" needs references.

15 References added

12. Page 4, Line 24: I believe boninitic basalts was also mentioned earlier? These need to be defined at first instance (i.e., lavas which record MgO >8 wt. % and TiO2 <0.5 wt. % as per IUGS boninite definition but SiO2 <52 wt. %)

Sentence modified

13. Page 5, Line 17: Again, confused as to whether paper included Coto Block; Maybe state in first sentence of this paragraph that study was conducted on Acoje Block (only) Sentence changed to "For this study, a subset of forty-four (44) samples located along NW-SE transects of the Acoje Block volcanic sequence at Barlo were selected for whole rock geochemical analyses and screened through visual and microscopic assessment of secondary

25 alteration."

14. Page 5, Line 24: change "lost weight"→ weight lost changed to "weight lost"
15. Page 5, Line 31: Spell out GSJ/AIST

now "Geological Survey of Japan/National Institute of Advanced Industrial Science and Technology"

16. Page 6: Section 5.1: This section is very "dense" and difficult to read. I suggest making a table showing the different lithologies and their mineralogy and textures, and then significantly shortening the written description here

Seems ok to us

5

17. Page 6, Line 15: Should of probably brought this up earlier, but similar to point 12 above, perhaps all the different categories of boninite (low-Si, high-Si, basaltic etc.) should be explained in the introduction or at least at an earlier point in ms

10 18. Page 6, Line 20: I think this is the first mention of Ogasawara? Mention where this is-Japan → part of IBM forearc?

Sentence changed to "This assemblage corresponds to Type II boninite of Umino (1986) in samples described from the type locality at Chichijima island, Ogasawara (Bonin) Archipelago in southern Japan."

15 19. Page 6, Line 24: change is → are changed
20. Page 7 Line 6: Insert "A" before peculiar changed

- 20 21. Page 7, Line 15: These LOI are very high. And what rock types exhibit these values? All or just high-Si boninites? These should be in the Table in Supp. Doc., correct? So put Supp. Doc. X at end of sentence. Maybe a plot of LOI versus various potentially mobile elements (e.g., MgO, K2O, Na2O, Ba) is warranted? Or at least some sort of statement like "though the LOI are high our petrologic arguments are based primarily on trace elements known to be
- 25 immobile up to greenschist-facies conditions". Are any filters being applied to your samples? For example, using only samples which yield 98-102 wt.% oxides or ones with <3 % LOI? Yes, indeed the LOI values are a bit high. We note that in general boninites have LOI values that are greater than 3 wt%.

IODP Expedition 352 shipboard boninite samples have LOI values that are much higher (mostly between 5-11 wt%) and samples from Haugen, 2017 which are the "cleanest, least weathered pieces" of the shipboard cores have LOI values that are just as much as the samples from this study.

5 Concerning element mobility, this is addressed in Page 7 lines 15-19.
Filters have already been applied in the current dataset. The current dataset was screened not only based on LOI values but primarily based on element immobility judged using diagrams such as CaO vs. Na2O, major oxide vs LOI, and major oxide/trace element vs. immobile element (e.g. Zr). Samples with mobile element enrichment (e.g. total alkalis, Rb) followed a
10 vertical vector and were excluded.

22. Page 7, Line 18: What is primary? And I note here that primary is used later on but not defined. Do you mean primary lavas such that exhibit high MgO, high Mg# (≅65), high Cr and high Ni?

Sentence modified.

15 23. Page 7, Line 22: I think a reference is needed after "boninitic basalts". Maybe Pearce and Robinson, or Reagan et al. (2017)? Not sure.

Reagan et al. (2015) added

24. Page 7, Line 23: As mentioned at the beginning, Figure 6 is very "busy". Would suggest plotting only samples from this study.

20 We slightly modified Figure 6. We removed objects that are not discussed in the text such the primary PAB and PAB fractionation. However, we've chosen to still include the Ogasawara and Expedition 352 boninite and boninite series datasets. Our point is to compare Zambales boninite with the dataset from the IBM forerac. Another point is to show the major element variation vis-à-vis the modelled fractional crystallization path. We believe these are better

25 addressed if our data are shown with the Ogasawara and Expedition 352 datasets. The panels are now marked as "a, b, c"

25. Page 7, Line 24: change second "and"  $\rightarrow$  or changed

26. Page 7, Line 29: pristine? You mention above LOI values of 4-7 wt.% Addressed fully in #21 27. Page 7, Line 31: ug/g? usually reported in ppm Indeed, trace elements are traditionally reported in ppm but as per Wiedenbeck, (2017) ug/g is 5 better 28. Page 8, Line 3: replace within with  $\rightarrow$  between changed to between 29. Page 8, Line 15: Change so reads: Compositions of Zambales boninite. . ... are marked by 10 low incompatible trace element abundances... changed to "Compositions of Zambales boninite and boninite series volcanics are marked by low incompatible trace element abundances relative to MORB" 30. Page 8, Line 23: replace times with  $\rightarrow x$ replaced with x 15 31. Page 8, Line 25: insert boninite between Zambales and ophiolite now "The effect of fractionation in Zambales boninite samples" 32. Page 8, Line 32: descending order is unclear; perhaps describe from base  $\rightarrow$  top which is probably standard convention now "The volcanic sequence at Barlo, based on observed local stratigraphic relationships, is 20 LSB series volcanics-boninitic basalt-LSB-HSB from base to top." 33. Page 9, Line 6: Is unclear how can be classified as moderate-Fe tholeiites without the Miyahsiro plot overlain by Arculus' low-med-high Fe series fields Figure will be added as Supplementary Figure 1 34. Page 9, Line 20: Haugen (2017): Is this a MSc or PhD thesis (not indicated in references). 25 References have been updated.

35. Page 10, Line 4: I think a paragraph at least is warranted to explain how the modelling was done using MELTS (supplementary document probably appropriate).

The modelling is fairly straightforward and the parameters are given in the figure captions. Figure 6 captions is now modified

36. Page 10, Line 5: Ghiorso and Gualda (2015) not in references Ghiorso and Gualda (2015) is in the references

5 37. Page 10, Line 16: change "in the base" to → at the base 5 change to "at the base"
38. Page 10, Line 17: break sentence; add ";" after "at depth" now "at depth; influx"

10 39. Page 10, Line 20: change slightly deviate  $\rightarrow$  deviate slightly changed to "deviate slightly"

40. Page 10, Line 24: change does  $\rightarrow$  do changed

41. Page 10, bottom of page: Would change Section 6.2 title to  $\rightarrow$  Slab contributions

15 now "6.2 Slab contributions to a less depleted mantle source"

42. Page 10, bottom of page: Would include more up-to-date references for boninite petrogenesis

additional references added

43. Page 11, Line 7-8: OK, but they are equally LREE-depleted. What is the explanation for

20 the spoon-shaped REE patterns? I think for the "classical" U-shaped signatures that the explanation is high-degree partial melting (which produces low MREE) which is subsequently slab-fluid modified to produce LREE enrichment; not sure of explanation for the high HREE less depleted source

44. Page 11, Line 13: Maybe explain at beginning of section that Ba/Th is a marker/gauge of

25 shallow slab-contributions and reference (Pearce et al., ?) Why Ba liberated at shallow conditions? Low temperature (I think); low P as well?

45. Page 11, Line 14: insert "increasing" after "mirrored by" now "mirrored by increasing"

46. Page 11, Line 15: reference Fig. 8b after Th/Yb Fig. 8b added

47. Page 11, Line 16: what is decoupled?

48. Page 11, Line 18: Insert "A" before "high U/Th ratio"

5 now "A high U/Th ratio"

49. Page 11, Line 19: change ratio ! ratios; change by ! if; add "to source" at end of sentence changed

50. Page 11, Line 21: the La/Th vs. Sm/La is not shown so have to indicate this "not shown" now indicated

10 51. Page 11, Lines 30-31: Why mention slab melts? This is not mentioned previously and I don't think anyone familiar with boninite petrogenesis would consider slab melts as part of the equation.

Sentence now modified

52. Page 11, Line 2: change so reads: transitional between MORB and IAT

15 changed

53. Page 12, Line 3: Change "in"  $\rightarrow$  on the basis of

changed

54. Page 12, Lines 3-4: Confusing sentence; why mention distinct from Mariana BAB?

20 Mariana BAB is mentioned because originally Hawkins and Evans, 1983 characterized Coto Block as back-arc basin oceanic crust. By comparing certain immobile elements (e.g Hf, Ta, Th), Geary et al., 1989 was able to show that it is distinct from Mariana BAB and that is has composition transitional to MORB and IAT.

55. Page 12, Line 5: Another confusing sentence; have to get point across that depletion in
REEs, TiO2, Zr and Y of Acoje relative to Coto documents the progressive depletion of. . .what about LILE enrichments? These should increase from Coto ! Acoje

LILE enrichments are not discussed for two simple reasons- (1) paucity of reliable LILE analyses in the compiled dataset (Hawkins and Evans, 1983; Geary et al., 1989; Yumul, 1990,

Evans et al., 1991, Tamayo, 2001), in fact no samples from the Coto Block crustal section have been analyzed by ICP-MS and (2) the altered nature of most samples, with most samples lacking petrographic characterization, makes it difficult to assess the primary nature of LILE enrichment.

- 5 56. Page 12, Line 10: Can't readily see where Coto lavas plot in Ti/V space but this is a very important point as FAB can be distinguished by MORB on the basis of Ti/V which is arc-like (>20) and by virtue of elevated Fe3+/FeT indicative of a more oxidized (arc-like) source. Suggest you state what the Ti/V ratios of the Coto lavas are and compare these with those of IBM. Are they similar or not?
- 10 We increased the symbol size of Coto volcanics in Fig 6.

Coto Block volcanics (n=9) have variable Ti/V ratios, mostly between 20-26 (Fig. 6f). Recognizing the small sample size, this range is higher than proto-arc basalts (PAB) and overlaps with MORB (Supplementary Fig. 2a); suggesting a less oxidized source compared to IBM PAB.

We also added the primitive mantle normalized immobile element pattern for Coto Block volcanics together with IBM PAB (Supplementary Figure 2b) to highlight its depleted nature.
57. Page 12, Section 6.4. See the Presentation and Scientific Interpretations section
58. Page 13, Line 12: insert "above a west-dipping subduction zone" after (Ishizuka et al., 2011)

20 changed

59. Page 13, Lines 15-20: sounds perhaps like a plume-induced SI scenario addressed in first-half of response

60. Page 14, Lines 6-8: Why feasible? No explanation for this (doubly-vergent subduction) addressed in first-half of response

8

25 61. Page 14, Line 11: Change north-verging → NE-verging changed to NE-verging

In addition, we slightly modified Fig. 7 because Nepoui and Cape Vogel HSB were mislabeled.

Fig. 7 is now updated

**Authors' reply to RC #2**

- 5 Page 1 Line 13: Delete "uniquely" and just say "predominant." We feel that it is indeed unique, especially the occurrence of high-silica boninite. This point is raised by Reagan et al., (2017) as well.
- Page 2 Lines 2-4: The statement that Cenozoic subduction initiation (SI) requires pre-existing weak zones or lithospheric collapse is erroneous. All models of SI require a weak zone, even lithospheric collapse.

This statement is a general characterization to differentiate spontaneous and induced SI from plume-induced SI.

Page 2 Lines 9-10: I wouldn't say that the IBM is "the most appropriate locality" to test models of subduction initiation. It might be the most appropriate place to test the Stern and Bloomer-type model, but there are other models as well. Maybe say "one of the most appropriate..." Sentence modified → "...one of the most appropriate..."

20

- Page 2 Lines 24-25: Cite references for "spontaneous" (e.g., Leng and Gurnis) and "induced" (e.g., Hall et al.) SI that have been replicated by numerical modeling. These references were cited in the following sentences (original MS page 2 lines 25-29.
- 25 Page 2 Line 28: delete semi-colon Semi-colon deleted.

Page 2 Line 30: Maybe change "unequivocally be attributed" to "uniquely explain." Sentence modified.

30

Page 3 Line 1: In discussing the basement complexes of the Philippines, Encarnacion (2004) might be an appropriate paper to cite. (Encarnacion, 2004, Multiple ophiolite generation preserved in the northern Philippines. Tectonophysics.) Additional references added.

9

35

Page 3 Line 17: Change "structurally bound" to "fault-bound" (if that is what is meant by "structurally bound"). Changed to "fault-bound"

| 5
10 | Page 3 Lines 21-22: In reference to "terrane docking," I presume the Zambales ophiolite is the terrane. But what is it docking with? (Indeed, what do you mean by "docking"?) Here terrane docking is used in a way similar to Pubellier et al., (2004). We mean that by the Pliocene its transport from equatorial positions to its present location probably have ceased. The identity of what Zambales ophiolite "docked with" is still unknown. It may be a composite of the following (1) an ocean basin that predates the South China Sea (pre-2016 there is consensus that this is proto-South China Sea), the vestiges of which are the Jurassic-Cretaceous cherts in the West Luzon Shear Zone of Karig, (1983) and complemented by recent radiolarian studies of Queaño et al., (2017a, 2017b) and (2) the northern extension of the Palawan microcontinental block (based on seismic data, Arfai et al., (2011) speculates that West Luzon Basin west of Zambales ophiolite is partly underlain by continental basement). | Field Code Changed |
|---------|--------------------------------------------------------------------------------------------------------------------------------------------------------------------------------------------------------------------------------------------------------------------------------------------------------------------------------------------------------------------------------------------------------------------------------------------------------------------------------------------------------------------------------------------------------------------------------------------------------------------------------------------------------------------------------------------------------------------------------------------------------------------------------------------------------------------------------------------------------------------------------------------------------------------------------------------------------------------------------------------------------------------------------------|--------------------|
| 15      | Page 3 Line 23: Change "in the westernmost margin" to "just west" of the ophiolite. The Mesozoic cherts are not found in the ophiolite itself, but in mélange-type shear zone material west of the ophiolite.
The limited distribution of these cherts maybe more aptly described with "in the westernmost margin" rather than "just west" of the ophiolite.                                                                                                                                                                                                                                                                                                                                                                                                                                                                                                                                                                                                                                                                      |                    |
| 20      | Page 3 Line 23-25: Besides Queano et al. 2017, you should also cite Hawkins and Evans (1983) who first described these cherts and Encarnacion (2004) who provides additional description of the shear zone.
Additional references added - Hawkins and Evans (1983), Karig et al., (1986) and Encarnacion (2004).                                                                                                                                                                                                                                                                                                                                                                                                                                                                                                                                                                                                                                                                                                                  |                    |
| 25      | Page 3 Line 24: The shear zone is not "buried;" it is well-exposed in several places.
Sentence modified                                                                                                                                                                                                                                                                                                                                                                                                                                                                                                                                                                                                                                                                                                                                                                                                                                                                                                                           |                    |
| 30      | Page 4 Lines 2-4: You might want to note that the idea that the San Antonio massif is a displaced block from the north is disputed by Encarnacion et al. (1999) for lack of convincing evidence.
I re-visited the Encarnacion et al., (1999) paper but I can't seem to find the lines that dispute idea that the San Antonio massif is a displaced block from the north.
In Encarnacion (2004) page 119, it is stated that "Except for some possible tectonic displacements of locally derived blocks (Yumul et al, 1998) the ophiolite is a coherent slab".                                                                                                                                                                                                                                                                                                                                                                                                                                                                   |                    |
| 35      | Page 4 Line 18: You may want to add that the arc affinity of the Acoje block is also consistent with radiogenic isotope data (Pb, Sr, and Nd), which indicate hydrous fluid enrichment (Encarnacion et al., 1999).
This reference is cited in the discussion section 6.2                                                                                                                                                                                                                                                                                                                                                                                                                                                                                                                                                                                                                                                                                                                                                          |                    |
| 40      | Page 4 Line 24: What rock types were the "bedding planes" measured on?
Bedding planes were measured primarily on pillow lavas and tuff breccias.                                                                                                                                                                                                                                                                                                                                                                                                                                                                                                                                                                                                                                                                                                                                                                                                                                                                                  |                    |
|         | 10                                                                                                                                                                                                                                                                                                                                                                                                                                                                                                                                                                                                                                                                                                                                                                                                                                                                                                                                                                                                                                   |                    |

Page 4 Line 26: Change to "... NW-plunging anticline just south." Seems ok to us. 5 Page 4 Line 28: What does "conjugate intrusive directions" mean? (Intruded into conjugate fractures sets?). Please clarify. (Dikes with) conjugate intrusive directions pertain to obliquely intruding dikes. Line 32: Change to "subaqeous fall-out deposits" (?) 10 Changed. Page 5 Lines 3-4: Do the terms "Strombolian to Hawaiian fire fountaining" apply to subaqeous/submarine volcanics? I would think not(?) Are these deposits in fact subaerial? If so, that would be a very important finding! It would imply that portions of the Zamabales 15 ophiolite in fact contain what might be actual subaerial island arc crust, which would substantially change the paleogeography of Luzon. Please clarify. Here we are referring to submarine Strombolian to Hawaiian-type fire fountaining. Sentence is modified and a reference discussing deep submarine pyroclastic eruptions (Head Field Code Changed and Wilson, 2003) is added. 20 Page 5 Line 12: What are the "upright structures"? Sedimentary/volcaniclastic bedding? Please clarify. "Upright structures" include flattened flow lobes in the summit of the recognized pillow volcano shown in Fig3d (inset). "Upright" is used in page 5 line 12 to describe the primary igneous structures (emplacementrelated) in the synclinal area in a broad sense. It is used in contrast to post-ophiolite emplacement folding affecting the volcanic section and overlying sedimentary sequences. Page 6 Line 1: Change "12" to "Twelve." (Avoid numerals at the beginning of a sentence.) Sentence modified. 30 Page 6 Line 2: Add "Standard JB-2 and JB-3 ..." Sentence modified. Page 7 Line 15: Change to "The Cs, Rb, ... and Mn contents in ..." 35 Sentence modified. Page 8 Line 16: Change to "...MORB for most trace elements..."

40

I think this is not necessary.

Page 10 Line 32: Please add a few words summarizing the general consensus on boninite genesis. ("...there is general consensus on boninite petrogenesis, namely that..."

Sentence modified.

Page 12 Line 3: Change semi-colon to a period. Sentence modified.

Line 7: Change "Ma" to "Myr." (You are referring to differences in ages, not a point in time.) Sentence modified.

Page 13 Lines 2-5: Regarding the paleomagnetic data, do the declinations support a
 Philippine Sea Plate connection as well? The Philippine Sea Plate has been inferred to have rotated clockwise. There are additional previous paleomagnetic studies (for example, Fuller et al., 1991, J. Asian Earth Sciences) that appear to demonstrate counterclockwise rotation for Luzon.

Declination-based interpretations of the Cenozoic rotation history of Luzon and Zambales are contentious; Fuller et al., (1991) argues for CCW rotation while McCabe et al., (1987) calls for CW rotation.

As discussed by the Queano et al., (2007), the use of declination data in tectonic models of Luzon and Zambales might be of limited use since the CW and CCW directions cannot be unambiguously ascribed to either local or major plate rotations.

20

5

Page 13 Line 20: By "rapid emplacement" do you mean "rapid formation"? If not, what do you mean by "rapid emplacement"?

We do not mean rapid formation. Rapid emplacement is inferred based on the reasoning that juvenile arc magmatism progressed only up to boninite; that it is a case of "failed' subduction initiation or arrested arc development. On the other hand, the IBM forearc which can be

described as a case of "successful" subduction initiation that produced an intra-oceanic arc with calc-alkaline lavas (41-35 Ma) after boninite (48-44 Ma) and proto-arc basalt (52-48 Ma).

Page 13 Line 20: Change to "The timing of the proposed subduction inititation..." 30 Sentence modified.

Page 13 Lines 21-35 to lines 1-3 of page 14: This section discusses the relationship between the Zambales ophiolite and Eocene (Angat) and Cretaceous ophiolites in the east Luzon area. The authors state that the similarity in ages of the Zambales and Angat ophiolites presented in Encarnacion et al. (1993) "does not necessarily prove and affinity" between the two. But

- 35 Encarnacion et al. (1993) "does not necessarily prove and affinity" between the two. But Encarnacion et al. (1993) (and Encarnacion et al., 1999, and Encarnacion, 2004) did not use the ages alone in arguing that the Zambales and Angat ophiolites (and Cretaceous ophiolites) are contiguous (not exotic/allochthonous to each other). The geology and stratigraphy as well are consistent with the Zambales-Angat ophiolite forming adjacent to the Cretaceous ophiolite
- 40 in the east, which was the main point of Encarnacion et al., 1993 (and reiterated/amplified in Encarnacion, 2004).

| 1 | - |
|---|---|
|   |   |
|   | - |

| - | Field Code Changed |
|---|--------------------|
|   | Field Code Changed |
|   | Field Code Changed |

|    | We disagree with the interpretations of Encarnacion et al. (1993).                                                                                                                                                                                                                                                                                                                                                                                                                                                                                                                          |       |                                          |
|----|---------------------------------------------------------------------------------------------------------------------------------------------------------------------------------------------------------------------------------------------------------------------------------------------------------------------------------------------------------------------------------------------------------------------------------------------------------------------------------------------------------------------------------------------------------------------------------------------|-------|------------------------------------------|
| 5  | In particular, we invite the reviewer to place the west dipping Cretaceous-Eocene subduction zone east of Sierra Madre that produced a late Cretaceous arc and an Eocene arc-backarc basin pair (presumably the Zambales ophiolite) shown in Fig.12 of Encarnación, (2004) in current global and regional plate reconstructions of SE Asia and the western Pacific region (Wu et al., 2016; Zahirovic et al., 2014, 2016).                                                                                                                                                                  |       | Field Code Changed Field Code Changed    |
| 10 | Indeed, age constraints were not the sole parameter used in arguing that the Zambales and Angat ophiolites (and Cretaceous ophiolites) are contiguous. But the Eocene volcaniclastic sequence, as shown in Fig.6 of Encarnación, (2004), overlies the Angat (Eocene) and Montalban (Cretaceous) ophiolites and not the Zambales ophiolite. Bachman et al., (1983) notes that there is marked lithelescal difference between the cast and weet flanks of the                                                                                                                                 | ***** | Field Code Changed
Field Code Changed |
|    | Luzon Central Valley Basin (CVB); he further characterized the CVB as a forearc basin.                                                                                                                                                                                                                                                                                                                                                                                                                                                                                                      |       |                                          |
| 20 | Regarding the issues mentioned above, I think suggesting that the Zambales is exotic to east Luzon causes greater problems with the proposed model, because the east Luzon ophiolites and arc crust are in-between the Zambales and the Philippines Sea Plate. In other words, if one wants to separate the Zambales ophiolite from east Luzon, shouldn't it also be separate from the Philippine Sea Plate?                                                                                                                                                                                |       |                                          |
| 25 | On the contrary, we find no problem if east Luzon ophiolites and arc crust are placed in-
between the Zambales ophiolite and the Philippine Sea Plate.
With regards to Zambales ophiolite being "separated" from the Philippines Sea Plate, this is
the case in earlier tectonic reconstructions which follows the arc- backarc basin scenario
where Zambales ophiolite is alternatively placed east of the Celebes Sea Basin in contact with
the western margin of the Philippine Sea Plate (Fig.5 of Rangin et al. , 1990a and Plate 2 of
Rangin et al., 1990b). |       | Field Code Changed
Field Code Changed |
| 30 | In its present form, the G-Plates based reconstruction in Fig 10b using the unfolded slab data, plate polygons and rotation file of Wu et al. (2016) is actually compatible with Encarnacion (2004) in a broad sense. If Zambales ophiolite was formed by the proposed incipient subduction on the western margin of the Philippine Sea Plate, one can argue that Zambales ophiolite can be continuous with basement of Luzon and that it is relatively autochthonous                                                                                                                       |       |                                          |
| 35 | (with respect to Luzon) (Encarnación, 2004); albeit highly displaced (relative to its present                                                                                                                                                                                                                                                                                                                                                                                                                                                                                               |       | Field Code Changed                       |
|    | position) caused by transport along a plate boundary (Pubellier et al., 2004). A back-arc origin for Eocene ophiolites associated with Cretaceous arc in eastern Philippines is compatible with Fig. 10b as well.                                                                                                                                                                                                                                                                                                                                                                           | ***** | Field Code Changed                       |
| 40 | Arguments against a backarc basin origin for Zambales are given in section 6.3; the only contention would then be the relationship of Angat to Zambales ophiolite.                                                                                                                                                                                                                                                                                                                                                                                                                          |       |                                          |

Page 14 Lines 5-7: Why is doubly-vergent subduction "feasible"? Please elaborate.

We speculate that the location of Philippine Sea Plate (PSP) in the nexus of Pacific, Indo-Australian and Eurasian plates and their long-term Cenozoic plate motion makes doublyvergent subduction initiation along its margins feasible. The northwestward translation and

5 clockwise rotation of the Philippine Sea Plate starting in the early Eocene had to be accommodated by the adjoining oceanic domain east of southern Eurasia (e.g. East Asian Sea); hence, its interaction with the oceanic leading edge of the Philippine Sea Plate is expected (Wu et al., 2016; Zahirovic et al., 2016) and likely led to incipient subduction (Fig. 10).

Page 14 Line 29: Change to "By studying the Zambales ophiolite..." Sentence modified.

Page 14 Line 29: Delete "(SI)" 15 Deleted.

Page 14 Line 29: It is stated that subduction initiation is a "plate-scale process." I'm not sure what the purpose of this statement is. When is it not a plate-scale process? Please clarify. The purpose of the statement is to emphasize that subduction initiation may not be localized in the purpose of the Philips of Plate Philips of Plate Plate and Plate Plat

- 20 the eastern margin of the Philippine Sea Plate. The doubly-vergent SI configuration presented here is distinct from current SI scenarios solely based on the IBM forearc which mainly focuses on the problem of whether subduction initiation is spontaneous or induced (e.g. Arculus et al., 2016; Keenan and Encarnación, 2016).
- 25 Figure 2: The caption should include the references for the ages shown in the figure. References added.

Figure 2: Page 5 says the 44.1 Ma age is from a sill. This isn't clear or indicated in the stratigraphic column.

14

30 It is shown in the stratigraphic column.

Figure 10: In panel "b", what are the diamonds, inverted triangles, and squares? Captions modified.

35

10

Field Code Changed

[revised manuscript text omitted]

Petrologic, geochemical and isotopic studies reveal that the Zambales ophiolite represents two distinct mantle-crust sequences - the 45.1 Ma Coto Block with affinity transitional to mid-ocean ridge basalt (MORB) and island arc tholeiite

30 (IAT) and the 44.2 - 43.7 Ma Acoje Block with island arc characteristics (Encarnación et al., 1999; Encarnación et al., 1993; Geary et al., 1989; Hawkins & Evans, 1983; Yumul, 1990). The existence of a NNE-SSW trending structural boundary previously characterized by Hawkins and Evans (1983) as a left-lateral fault between Acoje and Coto Blocks within the

17

| Deleted: structurally |  |
|-----------------------|--|
| Deleted: ed           |  |
| Deleted:              |  |

| Deleted: 2017a  |  |
|-----------------|--|
| Deleted: buried |  |
| Deleted: 2017b  |  |

| - | Deleted:          |
|---|-------------------|
| - | Deleted: affinity |

Masinloc massif is supported by gravity and magnetic data with anomaly contrasts which extend at depth (Salapare et al., 2015). Although the juvenile arc character of Acoje Block and the supra-subduction origin of Zambales ophiolite are clear, the crustal nature and tectonic environment of Acoje and Coto blocks remain unresolved. Acoje Block as defined by Yumul et al. (1990) consists of the northern Masinloc massif and the San Antonio massif while the Coto Block consists of the

- southern Masinloc massif and the Cabangan massif (Fig. 1b). The present disposition of Acoje Block is postulated to be the 5 result of southward translation of San Antonio massif with respect to northern Masinloc massif (Yumul et al., 1998). Crustal thickness estimates, based on SW-NE transects of the northern Masinloc massif, are up to 9.5 km for the mantle section and 7 km for the lower crustal section. The fertile to moderately depleted Acoje mantle section is comprised of harzburgite and lherzolite with spinel Cr# [Cr/(Cr+Al)] ranging from 0.18 to 0.56, generally increasing towards the mantle-crust transition
- zone (Evans and Hawkins, 1989; Yumul, 1990). Harzburgites from the uppermost mantle section are strongly depleted in light and middle rare earth elements (LREEs and MREEs) with equilibration temperature and oxygen fugacity estimates of 730° C and 1.9 Alog(fO2) units above the FMQ buffer, respectively (Tamayo, 2001), From the residual mantle section, the mantle-crust transition zone passes through interlayered ultramafic cumulates and dunite to layered cumulate gabbronorite. The Acoje transition zone hosts podiform chromite deposits (Cr# = 0.71-0.77) and Ni-Cu sulfides both of which are PGE-
- bearing (Bacuta et al., 1990). Refractory transition zone dunites, characterized by spinel Cr# greater than 0.6, are interpreted 15 to be of cumulate origin (Abrajano et al., 1989). The Acoje Block lower crustal section is dominated by cumulate gabbronorites with olivine-clinopyroxene-orthopyroxene-plagioclase crystallization sequence. Coexisting plagioclase and  $clinopyroxene \ in \ gabbronorite \ are \ calcic \ (anorthite \ content \ [Ca/(Ca+Na+K)] = 89-94) \ and \ magnesian \ (Mg\# \ [Mg/(Mg+Fe)] = 89-94) \ and \ magnesian \ (Mg\# \ [Mg/(Mg+Fe)] = 89-94) \ and \ magnesian \ (Mg\# \ [Mg/(Mg+Fe)] = 89-94) \ and \ magnesian \ (Mg\# \ [Mg/(Mg+Fe)] = 89-94) \ and \ magnesian \ (Mg\# \ [Mg/(Mg+Fe)] = 89-94) \ and \ magnesian \ (Mg\# \ [Mg/(Mg+Fe)] = 89-94) \ and \ magnesian \ (Mg\# \ [Mg/(Mg+Fe)] = 89-94) \ and \ magnesian \ (Mg\# \ [Mg/(Mg+Fe)] = 89-94) \ and \ magnesian \ (Mg\# \ [Mg/(Mg+Fe)] = 89-94) \ and \ magnesian \ (Mg\# \ [Mg/(Mg+Fe)] = 89-94) \ and \ magnesian \ (Mg\# \ [Mg/(Mg+Fe)] = 89-94) \ and \ magnesian \ (Mg\# \ [Mg/(Mg+Fe)] = 89-94) \ and \ magnesian \ (Mg\# \ [Mg/(Mg+Fe)] = 89-94) \ and \ magnesian \ (Mg\# \ [Mg/(Mg+Fe)] = 89-94) \ and \ magnesian \ (Mg\# \ [Mg/(Mg+Fe)] = 89-94) \ and \ (Mg\# \ [Mg/(Mg+Fe)]$ 0.80-0.87). In addition to petrological criteria discussed above, the arc affinity of Acoje Block has been demonstrated using
- 20 immobile element-based geochemical fingerprinting of volcanic and hypabyssal rocks, as well as constituent clinopyroxene (Yumul, 1996). On a regional scale, the ophiolite appears to have a domal structure with a north-south trending axis. The volcanic section of Acoje Block in northern Masinloc massif dips northwest and is unconformably overlain by middle to late Miocene sedimentary sequences

**3. Volcanic geology**

Based on observations around the Barlo massive sulfide mine, two volcanic units with distinct lithofacies are distinguished in the Acoje Block extrusive sequence-a dominantly pyroclastic lower basaltic andesite-andesite unit overlain by an upper boninite unit with minor boninitic basalts. Here and in the succeeding sections, we use the term boninite as per the recommended IUGS criterion (Le Maitre, 2002). Pillow lavas that qualify as boninite based on MgO and TiO2 contents but with < 52 wt% SiO2 are classified as boninitic basalts, equivalent to the basaltic boninite of Reagan et al., (2015). Structural

18

30

measurements of bedding planes in 136 locations constrain the gross structure of the crustal sequence, consisting of a NE-SW trending doubly-plunging anticline forming a dome structure and NW-plunging anticline (Fig. 2). Estimated thickness of the upper unit and lower volcanic units are 680 and 520 meters, respectively,

[revised manuscript text omitted]

**6.3 Subduction initiation origin of the Zambales ophiolite**

- The 45 Ma crustal section of the Coto Block has a composition transitional to mid-ocean ridge basalt (MORB) and island arc
- 10 tholeiite (IAT), On the basis of immobile element-based trace element discrimination diagrams, Coto Block dikes and lavas from Tarlac are robustly characterized as distinct from Mariana Trough back-arc basin basalts (Geary et al., 1989). Decreasing REEs, TiO2, Zr and Y in Coto volcanics, dikes in the Coto mantle section to Acoje volcanics document the progressive depletion of the mantle source in Zambales mantle wedge (Yumul, 1990). Succeeding zircon U-Pb geochronology established the 1–2 Myr, age difference between Coto and Acoje blocks (Encarnación et al., 1993). Thus,
- 15 combined geochemical and age constraints preclude a back-arc basin origin for Coto Block as forwarded by Hawkins and Evans, (1983), and are more compatible with a proto-forearc setting as originally suggested by Geary et al., (1989). Coto Block volcanics (n=9) have variable Ti/V ratios, mostly between 20-26 (Fig. 6f). Recognizing the small sample size, this range is higher than proto-arc basalts (PAB) and overlaps with MORB (Supplementary Fig. 2a); suggesting a less oxidized source compared to IBM PAB (Brounce et al., 2015; Hickey-Vargas et al., 2018; Mallmann and O'Neill, 2009). The
- 20 corresponding Ti8 and Si8 values of the Coto Block are comparable with proto-arc basalts from the Bonin Ridge, Mariana Trench slope, Hole U1430 of IODP Exp. 351 and Hole 1440 of IODP Exp. 352 (Arculus et al., 2015; Ishizuka et al., 2011; Reagan et al., 2010, 2015) (Fig. 9). Si8 and Ti8 values of Coto Block basalts are also distinct from Celebes Sea Basin and West Philippine Basin MORB (Pearce et al., 2005; Savov et al., 2006; Serri et al., 1991; Spadea et al., 1996; Zakariadze, 1981). In addition, the primitive mantle-normalized fluid immobile trace element abundances of Coto Block volcanics are
- 25 exceptionally low as well and more depleted compared to MORB (Hickey-Vargas et al., 2018) and Celebes Sea Basin basalts (Supplementary Fig. 2b). From observations in the field, oceanic accretion can be inferred from the Coto crustal section based on a well-developed sheeted dike complex, in contrast to the mutually intrusive dikes of the Acoje crustal section. With the recognition of boninite and boninite-series volcanics (Barlo), simultaneous and post-boninite moderate-Fe arc tholeiites (Barlo-Sual, Subic) in the crustal section of Acoje Block, it is apparent that the subduction initiation
- 30 chemostratigraphy (Ishizuka et al., 2011; Whattam & Stern, 2011) is present in the Zambales ophiolite as well. Although stratigraphic relationships of individual volcanic sections may be obfuscated by emplacement-related tectonic deformation, by considering available age constraints, the crustal sections of the Zambales ophiolite represent proto-arc accretion and

[revised manuscript text omitted]

- 20 discern that doubly-vergent SI geometry or subduction initiation with oppositely-dipping subduction zones along its margins is characteristic of 3-D thermo-mechanical models of plume-induced subduction initiation (Baes et al., 2016; Gerya et al., 2015). Plume-induced subduction initiation was first recognized by Whattam and Stern, (2015) to describe the temporal association of late Cretaceous plume-related oceanic plateaus in Central America that was followed by are volcanism with oppositely dipping slab dispositions. In the case of the Eocene Western Pacific, a mantle plume centered on the Manus Basin
- 25 had originally been invoked by Macpherson and Hall, (2001) to account for an inferred thermal anomaly in IBM boninite mantle source, implicitly connecting the initiation of the IBM arc to the presence of a mantle plume. Isotopic studies are in agreement that a mantle plume is likely associated with the oceanic plateaus and bathymetric highs emplaced at the same time as the opening of the West Philippine Basin (Hickey-Vargas et al., 2006; Ishizuka et al., 2013); however, thermal anomalies in excess of the ambient mantle are not reflected in mantle potential temperature estimates of proto-arc basalt
- 30 from the IBM forearc (Umino et al., 2017). A causative link between a mantle plume and the doubly-vergent SI configuration along Philippine Sea Plate margins is yet to be established. We speculate that the location of Philippine Sea Plate (PSP) in the nexus of Pacific, Indo-Australian and Eurasian plates and their long-term Cenozoic plate motion makes doubly-vergent subduction initiation along its margins feasible, The northwestward translation and clockwise rotation of the Philippine Sea Plate starting in the early Eocene had to be accommodated by the adjoining oceanic domain east of southerm

| 2 | c |
|---|---|
| 1 | 2 |
| - | 5 |

|
|------------------|
|                  |
|                  |
|
Delated (    |
|
|                  |
|                  |
|
|
|                  |
|                  |
|                  |
|                  |
|
|                  |
|                  |

Eurasia (e.g. East Asian Sea); hence, its interaction with the oceanic leading edge of the Philippine Sea Plate is expected (Wu et al., 2016; Zahirovic et al., 2016) and likely led to incipient subduction (Fig. 10). The dynamics of sustained doublevergent subduction is examined by Holt et al. (2017) but doubly-vergent subduction initiation is yet to be explored by numerical modelling. Field and petrologic data presented here demonstrate that models of subduction initiation based on the

- 5 IBM forearc are currently simplistic. Geodynamic models of subduction initiation should be based on robust geologic data and a doubly-vergent SI configuration based on the Zambales ophiolite provides another boundary condition for refined models of SI along PSP margins. We advocate that geodynamic models of subduction initiation along Philippine Sea Plate margins incorporate a pre-Eocene, N/NE-dipping, subduction zone (the proto\_West Philippine Trench of Faccenna et al., 2010) associated with Cretaceous terranes forming the overriding plate, doubly-vergent subduction initiation along its margins as well as the interplay of plate forces and mantle upwelling (e.g. Oki-Daito plume of Ishizuka et al. 2013) during
- 10 margins as well as the interplay of plate forces and mantle upwelling (e.g. Oki-Daito plume of Ishizuka et al. 2013) during incipient subduction (Fig. 10b).

**6** Conclusions**

[revised manuscript text omitted]

 Head, J. W. and Wilson, L.: Deep submarine pyroclastic eruptions: theory and predicted landforms and deposits, J. Volcanol. Geotherm. Res., 121(3–4), 155–193, doi:10.1016/S0377-0273(02)00425-0, 2003.
 Hickey-Vargas, R., Yogodzinski, G. M., Ishizuka, O., McCarthy, A., Bizimis, M., Kusano, Y., Savov, I. P. and Arculus, R.:

Origin of depleted basalts during subduction initiation and early development of the Izu-Bonin-Mariana island arc: Evidence from IODP expedition 351 site U1438, Amami-Sankaku basin, Geochim. Cosmochim. Acta, 229, 85–111, 10 doi:10.1016/J.GCA.2018.03.007, 2018.

Hickey-Vargas, R., Savov, I. P., Bizimis, M., Ishii, T. and Fujioka, K.: Origin of Diverse Geochemical Signatures in Igneous Rocks from the West Philippine Basin: Implications for Tectonic Models, pp. 287–303, American Geophysical Union., 2006.

[revised manuscript text omitted]

Macpherson, C. G. and Hall, R.: Tectonic setting of Eocene boninite magmatism in the Izu–Bonin–Mariana forearc, Earth
Planet. Sci. Lett., 186(2), 215–230, doi:10.1016/S0012-821X(01)00248-5, 2001.

Le Maitre, R. W.: Igneous Rocks. A Classification and Glossary of Terms. Recommendations of the International Union of Geological Sciences Subcommission on the Systematics of Igneous Rocks, edited by R. W. Le Maitre, Cambridge University Press., 2002.

Mallmann, G. and O'Neill, H. S. C.: The Crystal/Melt Partitioning of V during Mantle Melting as a Function of Oxygen
Fugacity Compared with some other Elements (Al, P, Ca, Sc, Ti, Cr, Fe, Ga, Y, Zr and Nb), J. Petrol., 50(9), 1765–1794, doi:10.1093/petrology/egp053, 2009.

[revised manuscript text omitted]